# The RepID–CRL4 ubiquitin ligase complex regulates metaphase to anaphase transition via BUB3 degradation

Sang-Min Jang[1], Jenny F. Nathans[2], Haiqing Fu[1], Christophe E. Redon[1], Lisa M. Jenkins [3], Bhushan L. Thakur[1], Lőrinc S. Pongor [1], Adrian M. Baris[1], Jacob M. Gross[1], Maura J. O'Neill[4], Fred E. Indig[5], Steven D. Cappell [2] & Mirit I. Aladjem[1]*

The spindle assembly checkpoint (SAC) prevents premature chromosome segregation by inactivating the anaphase promoting complex/cyclosome (APC/C) until all chromosomes are properly attached to mitotic spindles. Here we identify a role for Cullin–RING ubiquitin ligase complex 4 (CRL4), known for modulating DNA replication, as a crucial mitotic regulator that triggers the termination of the SAC and enables chromosome segregation. CRL4 is recruited to chromatin by the replication origin binding protein RepID/DCAF14/PHIP. During mitosis, CRL4 dissociates from RepID and replaces it with RB Binding Protein 7 (RBBP7), which ubiquitinates the SAC mediator BUB3 to enable mitotic exit. During interphase, BUB3 is protected from CRL4-mediated degradation by associating with promyelocytic leukemia (PML) nuclear bodies, ensuring its availability upon mitotic onset. Deficiencies in RepID, CRL4 or RBBP7 delay mitotic exit, increase genomic instability and enhance sensitivity to paclitaxel, a microtubule stabilizer and anti-tumor drug.

[1] Developmental Therapeutics Branch, Center for Cancer Research, National Cancer Institute, NIH, Bethesda, MD 20892-4255, USA. [2] Laboratory of Cancer Biology and Genetics, Center for Cancer Research, National Cancer Institute, NIH, Bethesda, MD 20892-4255, USA. [3] Laboratory of Cell Biology, Center for Cancer Research, National Cancer Institute, NIH, Bethesda, MD 20892-4255, USA. [4] Protein Characterization Laboratory, Cancer Research Technology Program, Frederick National Laboratory for Cancer Research, Frederick, MD 21701, USA. [5] Confocal Imaging Facility, National Institute on Aging, NIH, Baltimore, MD 21224, USA. *email: aladjemm@mail.nih.gov

Cell proliferation requires the continuous oscillation between the synthesis of DNA during interphase and the subsequent separation of chromosomes during mitosis. Transitions between these major events are orchestrated by posttranslational modifications to ensure the preservation of genomic information[1–4]. Ubiquitin ligases regulate cell cycle progression by mediating the timely degradation of effector proteins[5–8]. Cullin–RING ubiquitin ligase complex 4 (CRL4) associates with a diverse array of substrates through a series of WD motif containing substrate receptors known as DDB1-CUL4-associated factors (DCAFs)[9]. DCAFs mediate the recruitment of CRL4 to chromatin in two ways. First, DCAFs that contain a proliferating cell nuclear antigen (PCNA) interaction (PIP) domain, such as CDT2, recruit CRL4 to chromatin by associating with PCNA during DNA synthesis[10,11]. Second, another DCAF, replication initiation determinant protein (RepID)/PHIP/DCAF14, which contains a bromodomain[9] and a cryptic Tudor domain[12], can bind replication origins[13] and recruit CRL4 to chromatin prior to DNA replication[14]. During the G1–S transition of the cell cycle, CRL4 ubiquitinates CDT1, a licensing factor and member of the pre-replication complex (Fig. 1a, interphase)[3,9,10,15,16], to prevent excess replication origin licensing[17]. As DNA synthesis proceeds, CRL4 also ubiquitinates many other proteins, including Cyclin E[8], CDC6[18], and the replication factor MCM10[19]. After genome duplication is completed, another ubiquitin ligase, the anaphase-promoting complex/cyclosome (APC/C), targets Cyclin B1 and securin to enable a timely exit from mitosis and ensure that sister chromatids segregate equally to daughter cells[4,20–22].

Spindle assembly checkpoint (SAC) proteins (MAD1, MAD2, BUB1, BUBR1, and BUB3) preferentially associate with kinetochores and function as a surveillance network preventing premature chromosome segregation by blocking APC/C from associating with its coactivator, CDC20 (Fig. 1a, mitosis)[23,24]. Key components of the SAC include BUB1 and BUBR1, which form a complex (Mitotic Checkpoint Complex) with CDC20, and BUB3, which recruits BUB1/BUBR1 to the kinetochores[25–27]. After all chromosomes attach to microtubules, the Mitotic Checkpoint Complex dissociates from APC/C-CDC20, allowing CDC20 to activate APC/C[22,28–30]. Genetic disruption of SAC proteins is common in cancer, but complete inactivation of the SAC is lethal to normal and malignant cells alike, demonstrating that SAC function is essential for survival[31–33].

The triggering event that initiates the dissociation of SAC proteins, thereby enabling the transition from metaphase to anaphase, remains unclear. Surprisingly, we find that CRL4, which primarily is thought to regulate DNA replication and repair, plays a crucial role during mitosis by facilitating the ubiquitination of the SAC component BUB3, leading to the inactivation of the SAC and to the subsequent activation of APC/C and exit from mitosis. CRL4 is recruited to chromatin by the replication origin binding protein and metastatic melanoma marker RepID (DCAF14/PHIP)[13,34]. We find that, during mitosis, chromatin-bound CRL4 dissociates from RepID and binds

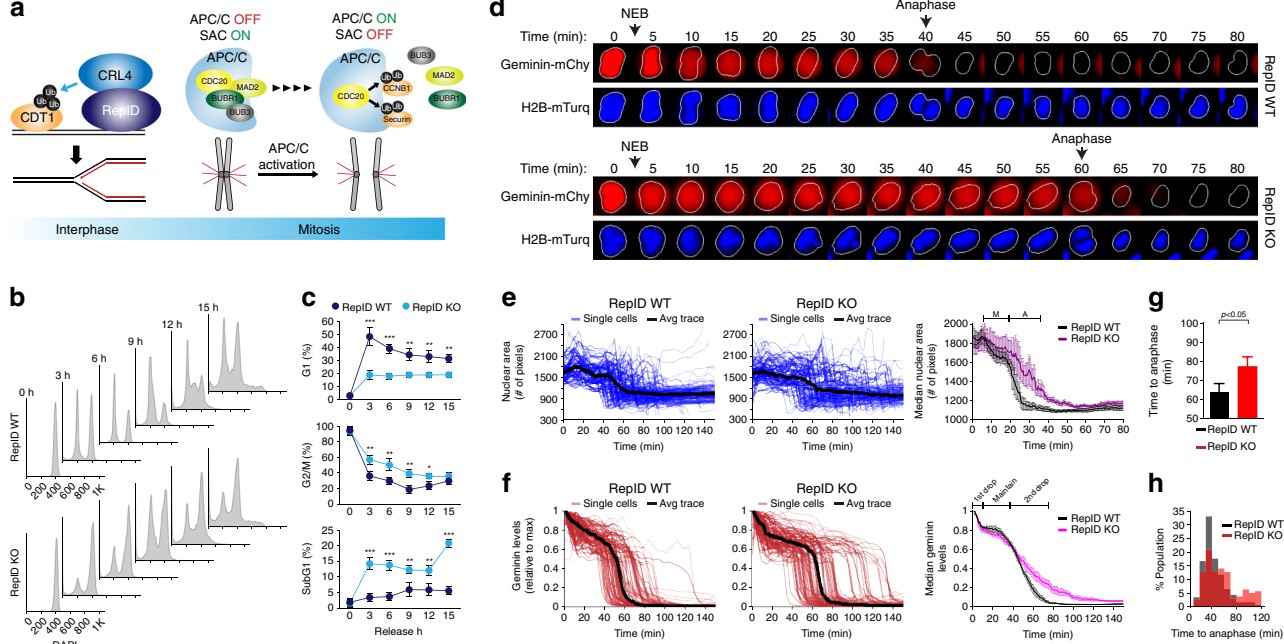

**Fig. 1 Role of RepID in mitotic exit and G1 entry. a** A model describing the current understanding of molecular interactions among components of the CRL4 and SAC complexes. **b**, **c** RepID KO cells delay mitotic exit. **b** HCT116 RepID WT and KO cells were exposed to nocodazole for 16 h, released into drug-free media, and collected every 3 h, followed by flow cytometry to monitor cell cycle progression. **c** Percentages of cells from **b** in G1, G2/M, and subG1 fractions for the experiment shown in **b**. Error bars in all results represent standard deviation from three independent experiments (*p value < 0.05, **p < 0.01, ***p < 0.001, Student's t test). **d–h** RepID KO cells exhibit prolonged metaphase–anaphase transition. **d** Image montage of a representative single cell expressing APC-degron (mCherry-geminin) and H2B-mTurquoise in HCT116 RepID WT and KO cells after release from CDK1 inhibitor-based synchronization. Images were taken every 5 min. NEB, nuclear envelop break. **e** Single-cell traces of the intensity of nuclear area in RepID WT and KO cells. The black line illustrates the average trace (left and middle panels). The first drop indicates a reduced area due to chromosome alignment in metaphase and the second drop indicates the segregation of chromosomes via the initiation of anaphase (right panel) (M metaphase, A anaphase). **f** Single-cell traces of APC-degron in RepID WT and KO cells. Black line illustrates the average trace (left and middle panels). The first drop indicates nuclear envelope breakdown (right panel). The constant APC-degron signal indicates a period prior to anaphase initiation. The second drop indicates anaphase initiation (right panel). **g** Bar graph indicates time to anaphase from release. **h** Percentage of anaphase cells in the population after release from nocodazole arrest in HCT116 RepID WT and KO cells.

another DCAF, tubulin-associated retinoblastoma binding protein 7 (RBBP7), which acts as a substrate receptor for BUB3. The CRL4[RBBP7] complex ubiquitinates kinetochore-associated BUB3, leading to its degradation and release of the SAC to allow mitotic exit. During interphase, BUB3 is protected from CRL4-mediated ubiquitination through its association with promyelocytic leukemia nuclear bodies (PML-NB). A reduction in RepID or CRL4[RBBP7] levels prevented ubiquitination of BUB3 and subsequently led to remarkably high cellular sensitivity to the microtubule stabilizer and antitumor drug paclitaxel (PTX), further suggesting the central role of CRL4 in mitotic exit. These observations provide insights into the role of CRL4 in mitosis, indicating that cells coordinate DNA replication and chromosome segregation by using the same ubiquitin ligase in different cell cycle phases. Our findings also illuminate the functional dynamics of DCAF switching and suggest that RepID levels could be investigated as possible effectors of cancer therapy.

## Results

**Role of RepID in mitotic exit and G1 entry**. To determine the chromatin-association dynamics of RepID during the cell cycle, we have arrested HCT116 cells in early mitosis by nocodazole, then released the cells into nocodazole-free medium and analyzed cell cycle progression. Surprisingly, we noticed that RepID-deficient (RepID knockout (KO)) cells[13] were significantly delayed in exiting mitosis and entering G1 phase as compared to RepID-expressing (RepID wild type (WT)) cells (Fig. 1b, c and Supplementary Fig. 1a). RepID-deficient cells also exhibited a significant increase in the prevalence of cleaved PARP1 (Supplementary Fig. 1b), concomitant with an increased subG1 (apoptotic) fraction (Fig. 1c), suggesting that a subpopulation of those cells undergoes apoptosis. In concordance, mitotic phosphorylation of histone H3 (pSer28) was not detected 3 h after release from nocodazole in RepID WT cells, whereas it was still detected up to 9 h after release from nocodazole in RepID-deficient cells (Supplementary Fig. 1b). These data indicate that RepID may play an unexpected role regulating mitotic exit.

To determine which mitotic phases were prolonged in RepID KO cells, we performed live-cell time-lapse microscopy and used the automated analysis software to track hundreds of single cells as they progressed through mitosis. We used cells (HCT116; RepID WT or KO) stably expressing histone H2B-mTurquoise, a chromatin marker, and an mCherry-conjugated APC/C degron motif of geminin, which detects APC/C activity[35,36]. Cells were arrested in the G2/M boundary using the cyclin-dependent kinase 1 (CDK1) inhibitor RO-3306, and upon removal of the inhibitor, the mitotic progression in single live cells was quantified by measuring the timing of nuclear envelope breakdown (NEB), the size of nuclear area, and the fluorescence intensity of the APC/C-degron reporter. In RepID WT cells, NEB was observed within 5 min after the removal of RO-3306. Geminin levels, an indicator of APC/C activation, disappeared about 40 min after release from RO-3306 arrest (Fig. 1d, upper panel). In RepID KO cells, the timing of NEB was identical, but geminin signals persisted (over 60 min—Fig. 1d, bottom panel and Supplementary Fig. 1c, d). Notably, CDK1-treated cells exhibited a shorter mitotic delay than nocodazole-treated cells, most likely because nocodazole can trigger a mid-prophase delay by inducing microtubule disassembly[37] and DNA damage[38]. The results from tracking hundreds of single live cells confirmed that the timing of NEB was similar in RepID WT and KO cells (Fig. 1e), whereas degradation of geminin was delayed in RepID KO cells, suggesting that APC/C activation was delayed in those cells (Fig. 1f). These observations indicated that RepID levels did not affect the progression between prometaphase and metaphase but

that RepID was required to advance the metaphase–anaphase transition (Fig. 1e–h and Supplementary Fig. 1c, d). In addition, the degradation of known APC/C substrates Cyclin B1 and Securin was also compromised in RepID KO cells (Supplementary Fig. 1e).

The metaphase–anaphase transition requires the dissociation of APC/C components from BUBR1, an SAC mediator (Fig. 1a)[28,29]. We thus examined the effect of RepID on the interaction between BUBR1 and members of the APC/C. We found that 30 min after release from nocodazole arrest, the interaction between BUBR1 and APC/C component APC4 was markedly decreased in RepID WT cells, whereas in RepID KO cells, the interaction between BUBR1 and APC/C members was prolonged (Supplementary Fig. 1f). The disappearance of the BUBR1–APC/C interaction was accompanied by the appearance of SUMOylated APC4 in RepID WT cells[20,21,39] but not in RepID KO cells (Supplementary Fig. 1f). These results suggest that the presence of RepID is required for the metaphase–anaphase transition.

**RepID-recruited CRL4 ubiquitinates BUB3 during mitosis**. To determine whether the RepID–CRL4 complex could interact with and modulate ubiquitination of substrates that are critical for mitotic progression, soluble nuclear and chromatin-bound fractions from U2OS cells stably expressing FLAG-RepID were immunoprecipitated with anti-FLAG-specific antibodies and analyzed by mass spectrometry (MS). One of the proteins identified by immunoprecipitation–MS (Fig. 2a) was BUB3, a subunit of the mitotic checkpoint complex and an APC/C E3 ligase inhibitor (Fig. 1a). Co-immunoprecipitation (co-IP) assays demonstrated that the WD40 domains of RepID were required for BUB3 binding (Fig. 2b). BUB3 interacted with CUL4 but not the other cullins (Fig. 2c, d), with the exception of a very weak interaction with CUL2. These observations suggested that CRL4 was the major Cullin–RING ubiquitin ligase able to bind BUB3. The interaction between BUB3 and RepID-CRL4 was restricted to G2/M phase (Fig. 2d).

Next we investigated whether RepID was required for the association of CRL4 with mitotic chromosomes. As previously reported[14], RepID forms a complex with CRL4 and recruits it to chromatin. Consistent with that notion (Fig. 3a), RepID-CRL4 localized to mitotic chromosomes in cells with intact RepID but not in cells deficient in RepID. Similarly, CUL4A/B and DDB1 were detected by immunoblotting in the chromatin fraction of cells with intact RepID but not in cells deficient in RepID (Fig. 3b). Notably, BUB3 localization to kinetochores during mitosis was not affected by RepID depletion (Fig. 3a). However, BUB3 protein levels markedly differed during cell cycle progression in RepID WT and KO cells. When we released RepID-proficient HCT116 cells from a nocodazole cell cycle block, which stalls cells in prometaphase, BUB3 levels decreased after mitosis, remained low during the G1 phase, and increased during the late S and G2 phases (Fig. 3c). In contrast, those levels remained high and constant in RepID-deficient cells subject to the same treatment and release from nocodazole block. Consistent with a role for RepID in BUB3 degradation during mitosis, the proteasome inhibitor MG132 prevented the decrease in BUB3 levels after mitotic release in RepID-proficient cells (Fig. 3d). These results suggested that RepID was essential for the recruitment of CRL4 to mitotic chromosomes and subsequently for the degradation of BUB3 by chromatin-bound CRL4.

We next tested whether RepID-CRL4 was directly involved in endogenous BUB3 ubiquitination by using a HIS-ubiquitin transfection assay. In RepID WT cells, BUB3 was ubiquitinated an hour after release from nocodazole, and this ubiquitination was observed in parallel with reduced BUB3 protein levels

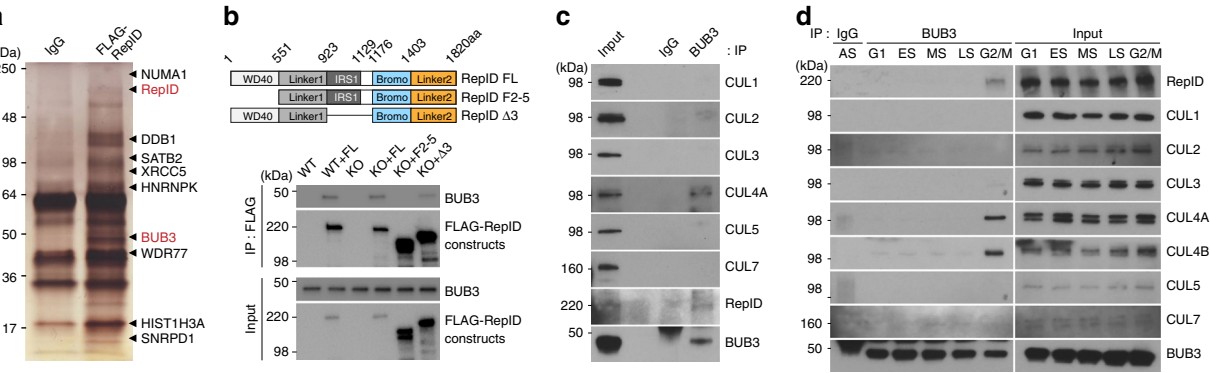

**Fig. 2 RepID-CRL4 interacts with BUB3 during mitosis. a** BUB3 interacts with RepID. Soluble nuclear and chromatin-bound fractions were extracted from RepID FL-stable U2OS cells and immunoprecipitated using an anti-FLAG antibody, followed by mass spectrometric analysis. **b** The WD40 domain of RepID is required for interaction with BUB3. Top, constructs of RepID variants used in this analysis. Bottom, nuclear and chromatin-bound fractions were extracted from U2OS cells expressing the RepID FL, F2–5, and Δ3 fragments and were immunoprecipitated with FLAG antibody and analyzed by immunoblotting. **c** BUB3 interacts with RepID-CRL4. Asynchronized HCT116 RepID WT cells were lysed, and BUB3-bound proteins were immunoprecipitated with anti-BUB3 antibody, followed by immunoblot analysis. **d** BUB3 interacts with RepID-CRL4 during mitosis. K562 RepID WT cells were fractionated by elutriation, and a mixture of soluble nuclear and chromatin-bound fractions were immunoprecipitated with anti-BUB3 antibody, followed by immunoblot analysis.

(Fig. 3e, lane 3). In contrast, ubiquitinated BUB3 was not detected in RepID KO cells even 3 h after release from nocodazole (Fig. 3e, lane 8). Similar results were obtained with two additional RepID-depleted cell lines (H1299, non-small cell lung cancer; DMS114, small cell lung cancer), which also exhibited low recruitment of CRL4 on chromatin following RepID depletion (Supplementary Fig. 2b–f). Notably, the fraction of cells with re-replicated DNA (>G2/M DNA content) was remarkably high in RepID-depleted DMS114 cells (Supplementary Fig. 2d). In all the cell lines tested, the re-introduction of the full-length (FL) RepID into RepID KO cells restored both the progression to G1 phase and BUB3 ubiquitination (Supplementary Fig. 2d–f).

To test whether CRL4 was involved directly in mediating RepID-facilitated BUB3 ubiquitination, we performed a HIS-ubiquitin assay in RepID WT cells transfected with small interfering RNAs (siRNAs) directed against CUL4A+4B. Depletion of both CUL4A and CUL4B prevented BUB3 from ubiquitination (Fig. 3f). Together, these results show that BUB3 is a substrate of CRL4, specifically during mitotic progression.

**RBBP7 is incorporated into CRL4 after dissociation of RepID**. We next asked whether RepID could act directly as a substrate receptor to ubiquitinate BUB3. Co-IP experiments showed that the association between RepID and CRL4 was reduced after mitotic release, suggesting that a large fraction of the RepID population dissociated from CRL4 after mitotic release (Fig. 4a). Dissociation of RepID from CRL4 preceded BUB3 degradation (Fig. 4a, input), suggesting that the degradation of BUB3 was mediated by another substrate receptor. To test this hypothesis, we immunoprecipitated CRL4 complexes using an antibody directed against the CRL4 component DDB1 in protein lysates from cells released from mitotic block and analyzed these extracts for the presence of known DCAFs[40,41]. The WD40-containing DCAF RBBP7, but not VprBP and RBBP4, was immunoprecipitated with FLAG-DDB1 concomitant with and immediately after the dissociation of RepID (Fig. 4b). Incorporation of RBBP7 into CRL4 after mitotic release was accompanied by an increased fraction of chromatin-bound RBBP7 (Fig. 4b, right panel). RBBP7 did not directly bind to RepID (Supplementary Fig. 3a). The interaction between BUB3 or CRL4 complexes and RBBP7 after mitotic release increased after RepID dissociation from CRL4, whereas RBBP7 binding to CRL4 was reduced in the absence of

RepID (Supplementary Fig. 3b). These observations support a model suggesting that RepID helps the handover of RBBP7 to CRL4. In vitro binding analyses demonstrated that both CRL4^RepID and CRL4^RBBP7 can interact with BUB3 (Supplementary Fig. 3c).

Confocal microscopic analysis revealed that RBBP7 associated with the mitotic spindle during mitosis, whereas other DCAFs known to interact with CRL4 (CDT2, VprBP, and RBBP4) did not localize to the mitotic spindle and were absent from mitotic chromosomes (Fig. 4c and Supplementary Fig. 3d, e). Super-resolution microscopy showed that RBBP7 could associate with BUB3-containing kinetochores, but BUB3-RBBP7 colocalization events were more evident when cells were treated with a p97/VCP inhibitor (Fig. 4d). These observations suggest that the association between BUB3 and RBBP7 at kinetochores occurred in a transient fashion during normal mitosis progression, immediately preceding the degradation of BUB3. Consistent with the above, colocalization between RBBP7 and CUL4A in RepID WT cells also increased in metaphase (Supplementary Fig. 3f).

To test whether the effects of RepID on mitotic exit were mediated by CRL4^RBBP7, we asked whether the mitotic exit phenotype in RepID depletion could be mimicked in RepID-proficient cells by depleting CRL4 subunits. Depletion of CUL4, DDB1, or RBBP7, or overexpression of BUB3 (BUB3 O/E), in RepID-proficient cells did not affect mitotic synchronization by nocodazole but did increase the mitotic fractions after release, suggesting that these conditions did not affect the entry to mitosis but delayed mitotic exit (Fig. 4e and Supplementary Fig. 3g). Depletion of CUL4, DDB1, and RBBP7 also facilitated apoptosis in a fraction of the cell population (Fig. 4e, subG1 fraction). In contrast, depletion of SKP2, a CRL1/SCF subunit, VprBP, or RBBP4 did not affect the cell cycle distribution in a similar manner (Fig. 4e and Supplementary Fig. 3g). Co-depletion of RBBP7/CUL4 or RBBP7/RepID did not exhibit synergistic or additive effects when compared to the phenotype of RBBP7- or RepID-depleted cells, suggesting that RepID, CRL4, and RBBP7 act in the same pathway and that RBBP7 acts downstream of CRL4 and RepID (Supplementary Fig. 3h). In agreement, depletion of RBBP7 (but not RBBP4) prevented the ubiquitination of BUB3 in RepID-proficient cells (Fig. 4f). In vitro ubiquitination assays performed with reconstituted purified components directly confirmed that BUB3 ubiquitination required the inclusion of both RBBP7 and CRL4 in the

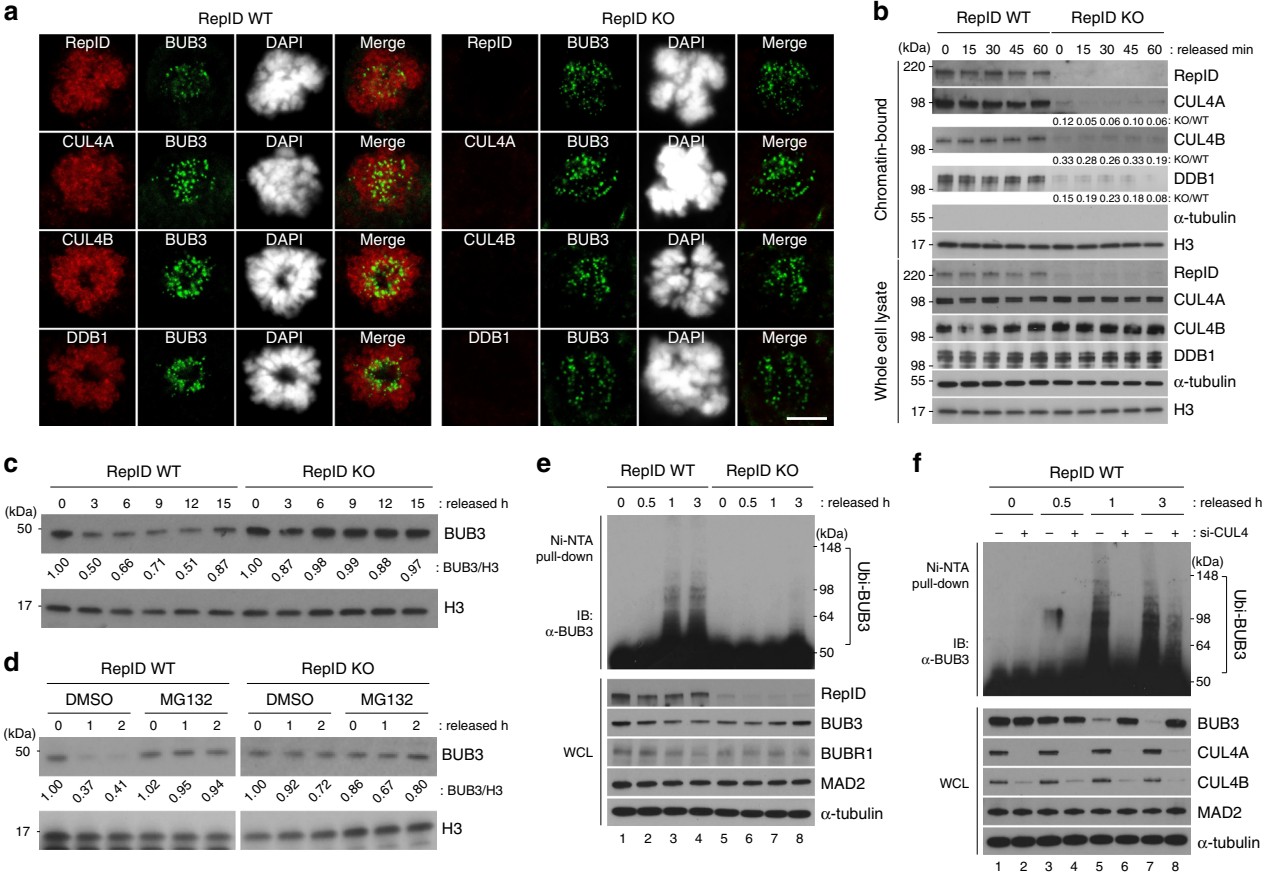

**Fig. 3 RepID-CRL4 ubiquitinates BUB3 during mitosis. a** RepID-CRL4 localizes with BUB3 at the kinetochore in prometaphase cells. Immunofluorescence analysis of HCT116 RepID WT and KO cells was carried out in methanol-fixed cells using antibodies directed against CRL4 components (RepID, CUL4A, CUL4B, or DDB1; red signals) and BUB3 (green signal). CRL4 components do not associate with chromatin in the RepID KO cells because RepID, which recruits CRL4 to chromatin, is absent. Scale bar: 10 μm. **b** RepID-dependent CRL4 recruitment to chromatin throughout the cell cycle. Mitotic HCT116 RepID WT or KO cells were collected after a nocodazole block, reseeded in drug-free medium, harvested at the indicated time points, and chromatin-bound proteins were analyzed by immunoblotting and densitometry. Histone H3 and α-Tubulin were used as a loading control. **c**, **d** BUB3 is degraded in mitosis in RepID-expressing cells but not in RepID-deficient cells. HCT116 RepID WT or KO cells were released from mitotic block for the indicated time periods with or without the proteasome inhibitor MG132. Cell lysates were subjected to immunoblot analysis and densitometry. **e** BUB3 is ubiquitinated during mitosis in RepID-expressing cells but not in RepID-deficient cells. HIS-tagged ubiquitin plasmids were transiently transfected into HCT116 RepID WT and KO cells. Cells were collected at the indicated times after release from a nocodazole block, and his-tagged proteins from the released cells were isolated on Ni-NTA beads followed by immunoblot analysis. **f** Knockdown of CUL4 prevented the ubiquitination of BUB3 in RepID-expressing cells. HIS-ubiquitin plasmid and siRNA-CUL4 (directed against both CUL4A and CUL4B) were co-transfected into HCT116 RepID WT cells, which were synchronized, released, lysed, isolated with Ni-NTA beads, and analyzed for the presence of ubiquitinated BUB3 by immunoblotting as in **e**.

reaction (Supplementary Fig. 3i). These observations suggest that RepID, a DCAF that recruits CRL4 to chromatin, dissociates from CRL4 and frees it to associate with RBBP7, a catalytic DCAF that ubiquitinates BUB3 during mitosis (Fig. 4g). These results imply that proper and timely mitotic progression entails the dissociation of RepID from chromatin-bound CRL4 and its replacement by RBBP7, which facilitates BUB3 degradation.

**RepID deficiency increases sensitivity to PTX.** PTX, a microtubule stabilizer, is a member of the taxane class of anticancer drugs used to treat many forms of cancer[42]. Because reduced expression of SAC proteins such as BUBR1 or MAD2 is associated with acquired PTX resistance[43], we examined the sensitivity of RepID WT and RepID KO cells to PTX. Cells with intact RepID were resistant to 5 nM PTX, whereas RepID-deficient cells exposed to the same dose of PTX for 24 h exhibited an increased mitotic fraction, marked re-replication of genomic DNA (5-

ethynyl-2′-deoxyuridine (EdU)-positive cells with DNA content of >4N) and higher levels of phosphorylated histone H3 (Fig. 5a, b). RepID-depleted cells exposed for 24 h to PTX showed a modestly higher fraction of subG1 (apoptotic) cell populations than RepID-expressing cells (5 nM: 8.25% vs. 5.16% in KO and WT, respectively; 10 nM: 23.4% vs. 16.9% in KO and WT; Fig. 5a, b). Cells expressing intact RepID were also refractory to 1 nM PTX, whereas RepID-deficient cells showed fewer surviving colonies after exposure to 1 nM PTX (Fig. 5c). In contrast, cell growth assays showed that RepID WT and KO cells exhibit similar sensitivities to the microtubule polymerization inhibitor nocodazole (Fig. 5d). The acute apoptosis we have observed in RepID-deficient cells in short-term assays (Fig. 1c and Supplementary Fig. 1a) did not translate to a significant loss of viability in long-term assays, most likely due to the lower dose used in colony-formation studies.

Exposure of RepID WT cells with either CRL4^RBBP7 depletion (si-CUL4, si-DDB1, si-RBBP7) or BUB3 O/E to 5 nM PTX increased the subG1 and mitotic fractions as well as the fraction

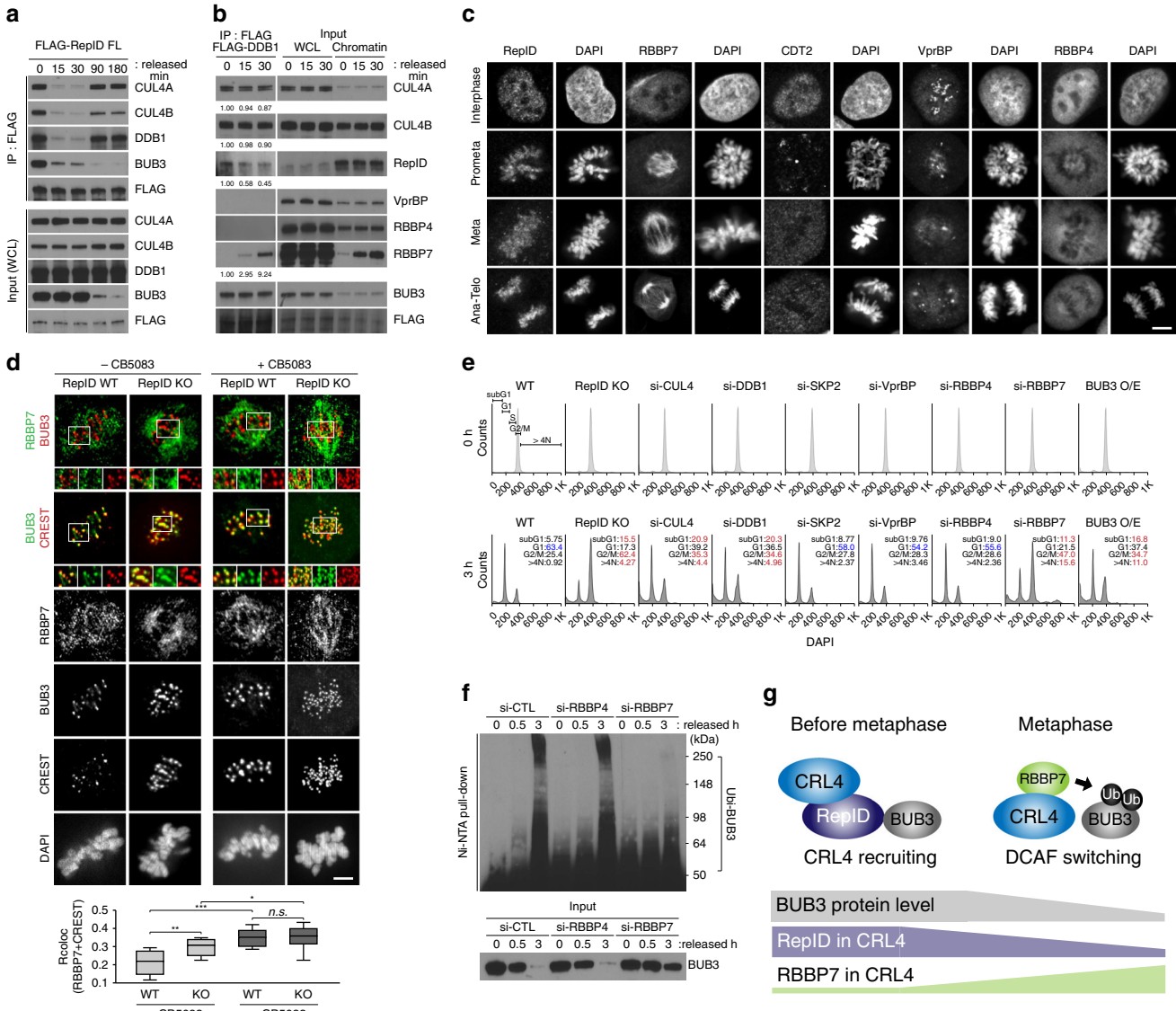

**Fig. 4 RBBP7 is incorporated into CRL4 and facilitates BUB3 ubiquitination. a** RepID dissociates from CRL4 during mitotic progression. HCT116 cells were transfected with RepID FL expression plasmid (for constructs, see Fig. 2b). Cells were collected at the indicated times after release from a nocodazole block, and chromatin-bound fractions were extracted. Immunoprecipitation was performed using anti-FLAG antibody, and co-precipitated proteins were analyzed by immunoblotting. **b** RBBP7 is incorporated into CRL4 after mitosis. Chromatin fractions of FLAG-DDB1-transfected HCT116 cells released from mitotic block at the indicated time points were immunoprecipitated with anti-FLAG antibody, followed by immunoblot analysis. **c** RBBP7, but not other DCAFs, localizes to mitotic spindle during mitosis. Immunofluorescence analysis was performed using the indicated anti-DCAFs antibodies in HCT116 WT cells. Scale bar: 10 μm. **d** RBBP7 colocalizes with BUB3 during metaphase. HCT116 RepID WT or KO cells were subject to immunofluorescence analysis using anti-RBBP7, BUB3, and anti-CREST antibodies with/without treatment with a p97/VCP inhibitor (CB5083). Zoomed-in areas (marked by white squares) are shown at the bottom of the composite images (merged images of the zoomed-in area are shown to the left of the single-color zoomed-in images). Scale bar: 10 μm. Quantification of colocalization between RBBP7 and BUB3 by Pearson's correlation coefficients (bottom panel) (*$p$ value < 0.05, **$p$ < 0.01, ***$p$ < 0.001, n.s. not significant, Student's $t$ test). **e** Depletion of RepID-CRL4$^{RBBP7}$ components or BUB3 overexpression can delay mitotic exit. Depleted or overexpressed HCT116 cells were released from a nocodazole arrest for 3 h and analyzed by flow cytometry. Percentages of cells in subG1, G1, G2/M, and >4N phase are indicated. **f** RBBP7 facilitates the ubiquitination of BUB3 during mitosis. HIS-ubiquitin plasmids were transfected into HCT116 RepID WT cells together with siRNA-RBBP4 or si-RBBP7. Cells were synchronized, released, lysed, and proteins isolated on Ni-NTA beads as described in the legend for Fig. 3e. Immunoblot analysis was carried out using anti-BUB3 antibodies. **g** A model describing the DCAF switch on chromatin during mitosis. RepID recruits CRL4 on chromatin before metaphase, bringing CRL4 in contact with BUB3. RepID then dissociates from CRL4, and RBBP7 is incorporated into CRL4 as a catalytic DCAF for BUB3 ubiquitination.

of >4N, while also causing a decrease in the G1 fraction (Fig. 5e, f). In contrast, depletion of SKP2 or RBBP4 did not result in a significantly different cell cycle distribution (Fig. 5e, f). RepID KO cells, CRL4$^{RBBP7}$-depleted cells, and cells with overexpressed BUB3 showed elevated multinucleation and micronuclei

following PTX treatment (Fig. 5g, h). Depletion of SKP2 or RBBP4 seemed to exhibit a similar level of resistance to PTX as WT cells (Fig. 5g, h). Together, these results suggest that failure of RepID-CRL4$^{RBBP7}$-dependent BUB3 degradation contributes to increased sensitivity to PTX.

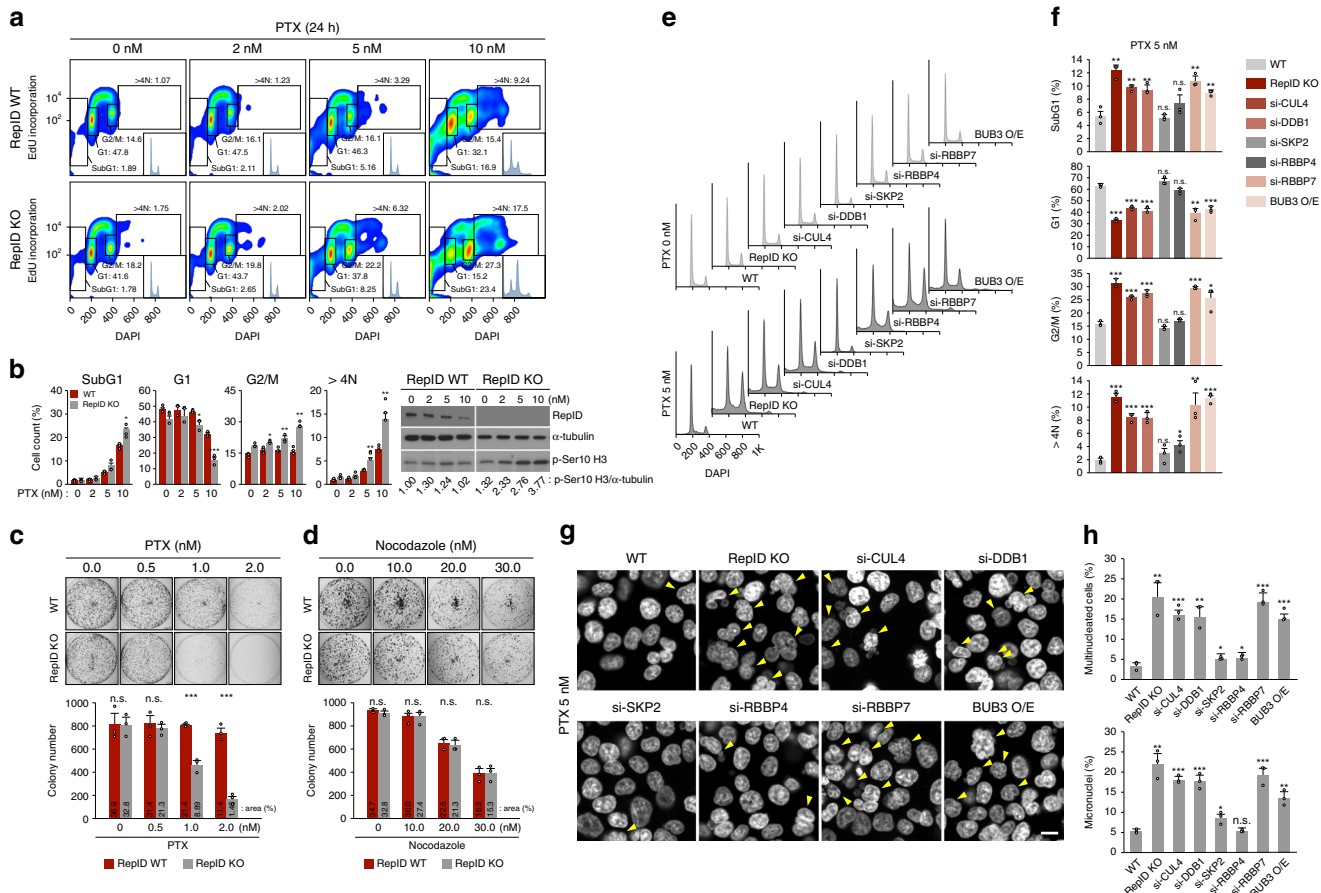

**Fig. 5 RepID-deficient cells are more sensitive to paclitaxel. a** Paclitaxel (PTX) treatment increases the prevalence of subG1, G2/M, and >4N populations in RepID-deficient HCT116 cells. HCT116 RepID WT and KO cells treated with PTX for 24 h were labeled with EdU for 30 min prior to collection and analyzed by FACS. Percentage of each cell cycle phase was indicated in EdU-positive histogram plots (insets). **b** Bar graph depicting the cell cycle distribution of cells collected in **a** from three biologically independent experiments (left panel). Immunoblot analysis showing elevated accumulation of phosphorylated histone H3 in RepID-deficient cells in response to PTX treatment as in **a** (right panel). **c** Colony-formation assay with HCT116 RepID WT and KO cells upon PTX treatment. Bar charts indicate colony number, and the percentage of covered area is provided above the bars. **d** Nocodazole sensitivity is not changed by RepID deficiency. **e**–**h** Failure of RepID-CRL4$^{RBBP7}$-based BUB3 degradation leads to increased sensitivity to PTX treatment. **e** Depleted/overexpressed HCT116 cells as indicated were treated with PTX and analyzed by FACS. Histogram indicates asynchronized (upper panel) or PTX-treated cells (bottom panel). **f** Percentage of cells in subG1, G1, G2/M, and >4N phase in **e**. **g** Nuclear staining after PTX treatment in depleted/overexpressed HCT116 cells as indicated. Red arrows indicate multinucleated cells. Scale bar: 10 μm. **h** Percentage of multinucleated cells and micronuclei in **g**. Error bars in all results represent standard deviation from three independent experiments (*$p$ value < 0.05, **$p$ < 0.01, ***$p$ < 0.001, n.s. not significant, Student's $t$ test).

## BUB3 colocalizes with PML-NB during interphase.

The above observations suggest that BUB3 is a substrate of CRL4 and that its interaction with CRL4 facilitates its ubiquitination and degradation during mitosis. Since we observed that BUB3 levels were stable, and even increased, during interphase, we sought to identify a mechanism that would prevent BUB3 degradation during interphase despite the presence of active, chromatin-bound CRL4[5,14]. As shown in Fig. 6a and Supplementary Fig. 4a, chromatin-bound BUB3 was detected in intense foci that colocalized with PML-NB. The presence of BUB3 in PML-NB was confirmed by three-dimensional analyses of Z-stacks imaged by super-resolution microscopy (Fig. 6b, c and Supplementary Fig. 4b). BUB3 was not detected in RepID-expressing, PML-depleted cells, whereas in cells depleted of both RepID and PML, BUB3 was detected in diffuse patterns (Fig. 6d, e, Supplementary Fig. 4c–e). Exposure to MG132 prevented BUB3 degradation in PML-depleted, RepID-proficient cells (Fig. 6e, f, Supplementary Fig. 4c, bottom panel, and Supplementary Fig. 4f), and BUB3 degradation was not observed in PML-depleted, RepID-depleted cells regardless of whether these cells were exposed to MG132

(Fig. 6e, f). These observations suggested that BUB3 was protected from degradation in RepID WT cells through its sequestration in PML-NB during interphase.

The effects of MG132 on BUB3 levels and distribution in PML-depleted RepID WT cells suggest that BUB3 might be a potential substrate for CRL4-mediated ubiquitination during interphase, unless it is protected by PML-NB. Consistent with this suggestion, the colocalization between BUB3 and CUL4A partially increased with exposure to MG132 (Supplementary Fig. 4g, RepID WT panel). CUL4A and BUB3 did not show colocalization in RepID KO cells, regardless of PML levels or MG132 treatment (Supplementary Fig. 4g, RepID KO panel). BUB3 proteins contain a PIP-like motif that is highly conserved across species, suggesting that, in PML-depleted cells, BUB3 might be ubiquitinated by PCNA-based CRL4$^{CDT2}$ (Supplementary Fig. 4h). As shown in Supplementary Fig. 4i, co-IP experiments indeed confirmed that CRL4$^{CDT2}$ interacts with BUB3 on chromatin during S phase in PML-depleted cells. Direct ubiquitination assays verified that BUB3 was not ubiquitinated in the presence of PML-NB (Fig. 6g, lanes 1–4) but was

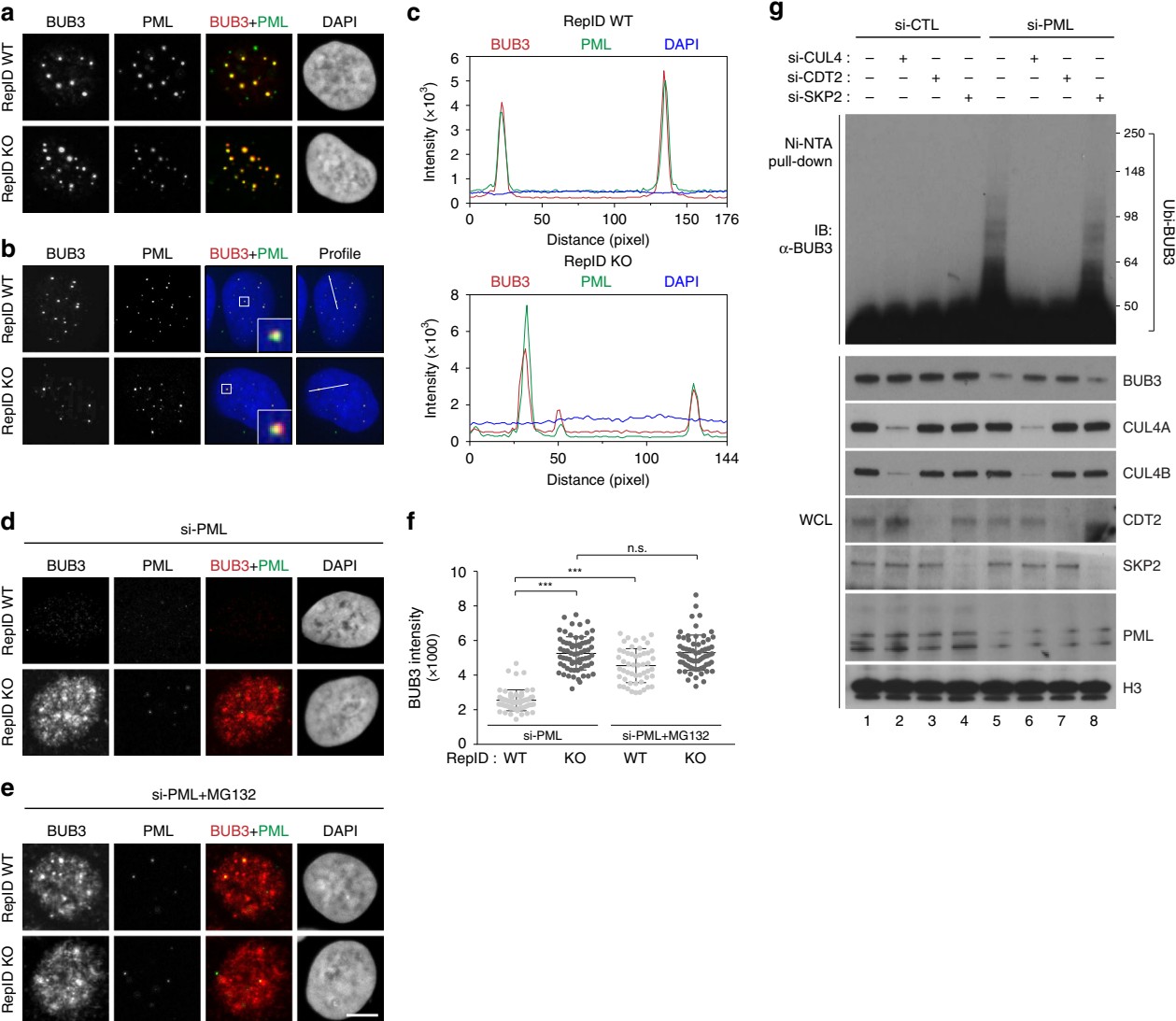

**Fig. 6 PML-NB protects BUB3 from CRL4$^{CDT2}$ during interphase. a–c** BUB3 colocalizes with PML-NB during interphase. U2OS RepID WT and KO cells were exposed to Triton X-100 to remove excess non-chromatin-bound proteins, and immunofluorescence analysis was performed using anti-PML and anti-BUB3 antibodies. Representative images using confocal microscopy (**a**) and super-resolution microscopy (**b**) are shown. **c** Colocalization between BUB3 and PML-NB was analyzed by intensity profiling as indicated in **b**. **d–f** BUB3 is degraded in RepID WT cells in which PML has been knocked down. siRNA-PML was transfected into U2OS RepID WT and KO cells, and immunofluorescence analysis was performed using anti-BUB3 and anti-PML antibodies. Representative images are shown without MG132 (**d**) or with MG132 (**e**). **f** Intensities of BUB3 in cells as in **d**, **e**. **g** BUB3 is ubiquitinated by CRL4$^{CDT2}$ in PML-deficient RepID WT cells. HIS-ubiquitin plasmid was transfected along with the indicated siRNAs into HCT116 RepID WT cells, and lysates were isolated on Ni-NTA beads, followed by immunoblot analysis. Error bars in all results represent standard deviation from three independent experiments (***$p$ value < 0.001, n.s.; not significant, Student's $t$ test).

ubiquitinated in PML-depleted cells (Fig. 6g, lane 5). Ubiquitination of BUB3 was prevented by depletion of either CUL4 or CDT2 but not by silencing of SKP2 (Fig. 6g, lanes 6–8). These results demonstrate that CRL4$^{CDT2}$ can target BUB3 during interphase, and this ubiquitination is prevented by association with PML-NB.

## Discussion
The cell cycle is driven by the combinatorial function of ubiquitin E3 ligases, which modulate the expression of effector proteins via ubiquitination-based proteasomal degradation. Previous studies have identified APC/C as the ubiquitin ligase driver of the metaphase–anaphase transition. The data presented here demonstrate that a second ubiquitin ligase, CRL4, is crucial for

proper mitotic exit. CRL4 facilitates APC/C activation via ubiquitination of BUB3, a subunit of the SAC that inhibits APC/C to prevent chromosome mis-segregation. These observations identify BUB3 as a substrate for CRL4.

Our findings suggest that BUB3 is protected from degradation by CRL4 during interphase by its association with PML-NB. Other instances in which the sequestration of proteins in PML-NB protects them from ubiquitination are known. For example, PML association plays a role in the inhibition of CRL3$^{KLHL20}$-mediated ubiquitination of death-associated protein kinase[44], and PML-NB facilitate the sequestration of the p53 ubiquitin E3 ligase MDM2 in nucleoli[45], which orchestrate protein trafficking to regulate protein availability during cellular stress[46,47]. The protection of BUB3 from degradation through association with PML-NB can provide a mechanistic basis for the observed role of

PML-NB in telomere stability[48], because BUB1–BUB3 complexes facilitate telomere DNA replication by directly phosphorylating telomere-associated TRF1[49].

CRL4 combines with numerous substrate receptors (DCAFs) to mediate ubiquitination. While associated with replicating chromatin, the activity of CRL4 on various substrates, including CDT1, p21$^{CIP1/WAF1}$, and SET8[9–11,50–52], requires an interaction of the substrates and the substrate receptor CDT2, a DCAF, with PCNA[5,11,18,53–55]. RepID is larger than other DCAFs and contains chromatin recognition domains (a Bromodomain and a Tudor domain) that can facilitate the direct recognition of chromatin[9,12]. We have recently reported that RepID recruits CRL4 to chromatin prior to DNA replication[14]. Combined with our current results, we propose that RepID is an example of a group of DCAFs, structural DCAFs. These structural DCAFs can actively recruit CRL4 to chromatin, in contrast to the other, catalytic DCAFs, which act as CRL4 substrate receptors once it is recruited to chromatin by other molecules (e.g., PCNA, which recruits CRL4 to chromatin during S phase). Specifically, RepID can recruit CRL4 to chromatin in G1 phase or in mitosis independent of PCNA.

We observed that, although RepID facilitated the recruitment of CRL4 to chromatin, it did not recruit BUB3 to kinetochores or RBBP7 to mitotic spindles. RBBP7 did not associate directly with RepID, and RepID dissociated from chromatin-bound CRL4 prior to BUB3 degradation. These observations suggest that RepID, the structural DCAF, recruits CRL4 and then is replaced within the CRL4 complex with RBBP7, the catalytic DCAF. The CRL4$^{RBBP7}$ complex is an active chromatin-bound ubiquitin ligase that recognizes the BUB3 as a substrate and facilitates its ubiquitination. CRL4$^{RBBP7}$-dependent ubiquitination of BUB3, associated with the mitotic checkpoint complex on mitotic chromosomes, releases the checkpoint by allowing CDC20 to bind to the APC/C and facilitates it activation (Fig. 7). This role of CRL4$^{RBBP7}$ is in a line with previous reports showing that CRL4$^{RBBP7}$-dependent ubiquitination affects CENP-A deposition at centromeres after mitotic exit[40]. Another study has shown that Aurora B kinase localization is controlled by ubiquitination mediated by CUL3/KLHL9/KLHL13, suggesting that progression of mitosis is governed by at least two types of ubiquitin E3 ligase[56]: CRLs and APC/C. The exact mechanism for DCAF switching during mitosis remains to be explored, and future studies are also required to address the question of how CRL4 is maintained on chromatin after transient dissociation of RepID.

Our model proposes that, during interphase, BUB3 is localized in PML-NB, which protects it from degradation (Fig. 7, interphase). Although RepID can recruit CRL4 to chromatin during interphase, and CRL4 can degrade substrates in association with the catalytic DCAF CDT2, BUB3's association with PML-NB protects it from CRL4$^{CDT2}$-mediated degradation. Hence, in RepID-proficient cells, the association of BUB3 with PML-NB maintains a pool of active BUB3 molecules, which further increases in the absence of CDT2 activity during the G2 phase of the cell cycle. This pool of protected BUB3 molecules is released from PML-NB at the correct time during mitosis, ready to interact with kinetochores and activate the SAC.

In RepID-deficient cells, CRL4 loading to mitotic chromosome is compromised, degradation of BUB3 is prevented because CRL4 is not associated with chromatin, and APC/C activation is delayed (Fig. 7, mitosis). Mitosis proceeds after a long delay in RepID-deficient cells, most likely through activation of the other cullin-anchored ubiquitin ligases, including CRL7 and CRL9[57,58]. The prolonged metaphase–anaphase transition we observe in RepID-depleted cells correlates with the increased sensitivity to PTX of cells depleted in RepID/CRL4$^{RBBP7}$ or overexpressing BUB3. This finding is in line with the observed PTX resistance of BUB1-depleted ovarian cancer cells[59] and in mice depleted of MAD2 and BUBR1[60]. The increased population of cells in subG1 phase observed after BUB3-overexpressing cells were released from nocodazole also implies that prolonged delays in mitotic exit might lead to cell death. These results are consistent with the failure of chromosome segregation and embryonic lethality in mice with homozygous depletions of MAD2, BUBR1 and BUB3, and with the mitotic problems and genomic instability observed following misregulation of BUB3 at either the protein or mRNA levels[61–65], analogous to overactivation and suppression of post-translational modifications in APC/C components[66,67].

PTX is used to treat many forms of cancer, and resistance to PTX is a significant health problem[68–70]. RepID is a marker and mediator of melanoma metastasis[34,71], and overexpression of RepID promotes the progression of a subset of melanoma, breast, and non-small cell lung cancers[71]. Our study provides a mechanistic basis for a role of RepID in cell proliferation and possibly in resistance to common therapies that modulate cell cycle progression.

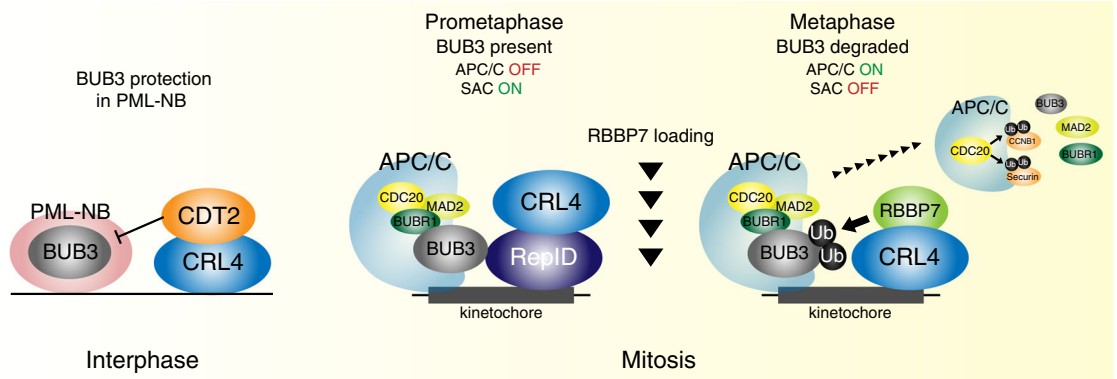

**Fig. 7 Schematic model.** RepID (structural DCAF) brings CRL4 on chromatin during interphase. BUB3 is localized to PML-NB, which protects BUB3 from degradation by CRL4$^{CDT2}$. In RepID-deficient cells, BUB3 degradation is compromised even when PML-NB levels are decreased because chromatin-bound CRL4 are low. During mitosis, RepID recruits CRL4 on chromosome and binds to the kinetochore-localizing BUB3. While RepID is dissociated from CRL4, mitotic spindle-localizing RBBP7 (catalytic DCAF) is incorporated into CRL4 during metaphase. CRL4$^{RBBP7}$-dependent degradation of BUB3 promotes dissociation of SAC, resulting in CDC20–APC/C interaction and anaphase initiation. RepID-deficient cells exhibit no changes in function, expression, or localization of RBBP7 and BUB3. Prolonged metaphase–anaphase transition by defective BUB3 degradation in RepID-deficient cells is due to a decrease of CRL4 on mitotic chromosomes, leading to increased sensitivity to PTX treatment.

## Methods

**Cell culture, chemicals, and synchronization**. Human U2OS, HCT116, and K562 cells with and without RepID were cultured in Dulbecco's modified Eagle's medium (Invitrogen, 10569-010) supplemented with 10% heat-inactivated fetal bovine serum and human H1299 and DMS114 cells with and without RepID were cultured in RPMI-1640 medium (Invitrogen, 11875–093) supplemented with 10% heat-inactivated fetal bovine serum in a 37 °C/5% $CO_2$ humidified incubator. All original cancer cell lines were obtained from ATCC (www.atcc.org), and all cell lines were tested as negative for mycoplasmas (Lonza, LT07–418). PTX (Sigma, T7191) was added to the media at the indicated concentrations. HCT116, H1299, and DMS114 cells were synchronized in prometaphase by a shake-off after 16 h of incubation in 100 nM nocodazole. Mitotic cells were washed three times in phosphate-buffered saline (PBS) and either collected immediately (0 h) or released in drug-free medium at various time periods. HCT116 cells were synchronized at the G1/S boundary using double thymidine block and released into fresh medium. The cell cycle distribution of the cells was confirmed by flow cytometry using an LSR Fortessa cell analyzer (BD Biosciences) after staining the DNA with 4′,6-diamidino-2-phenylindole (DAPI).

**RepID-depleted cell lines and transfection**. RepID was depleted in U2OS, HCT116, H1299, and DMS114 cells using CRISPR-CAS9. A 20-base pair guide sequence targeting the fifth exon of RepID (5′-CTGCAAATATGTCATCGACTAGG-3′) and the eighth exon of RepID (5′-GTGATAAAATGATCCGAGTCTGG-3′) in U2OS, HCT116, H1299, and DMS114 cells was selected from a published database of predicted high-specificity protospacer-PAM target sites in the human exome. Cells were cultured in 6-well dishes to 70–80% confluency for co-transfection with 2 µg of the RepID single guide RNA plasmid, 2 µg of linearized pCR2.1 vector harboring a puromycin-resistance gene expression cassette, and 10 µl of Lipofectamine 2000 (Life Technologies) per well. Cloning, selection, and verification were performed using PCR. Human FLAG-tagged expression plasmids for CUL4A (RC214798), CUL4B (RC206935), DDB1 (RC208372), BUB3 (RC200376), and RBBP7 (RC231256) were purchased from ORIGENE. The human siRNA oligo Duplex for DDB1 was purchased from ORIGENE (SR301160). All SMARTpool ON-TARGETplus siRNA were purchased from Dharmacon: CUL4A (L-012610), CUL4B (L-017965), PML (L-006547), CDT2 (L-020543), SKP2 (L-003324), VprBP (L-062261), RBBP4 (L-012137), RBBP7 (L-011375), and negative control siRNA (D-001810). siRNA transfection was performed using Lipofectamine RNAiMax (Invitrogen, 13778030).

**Constructs**. CSII-pEF-H2B-mTurquoise[72] and CSII-pEF-Geminin(aa1–110)-mCherry[35] were described previously and were used to make lentivirus. Stable HCT116 WT and HCT116 RepID KO cell lines expressing fluorescent reporters were generated using lentivirus. Transduced cells were sorted using a BD FACS Aria Fusion to obtain populations expressing the desired reporters.

**Flow cytometry analysis**. Cells were pulse-labeled with 10 µM EdU for 30 min prior to harvesting as well as with EdU staining using the Click-iT EdU Kit (Invitrogen, C10424). Staining was performed according to the manufacturer's protocol. DAPI was used for DNA counterstaining. An LSR Fortessa cell analyzer (BD Biosciences) with the FlowJo 10.5.2 software was used for cell cycle analyses. All experiments report representative results of at least three independent repetitions.

**Clonogenic survival assay**. Cells were plated in 6-well plates (500 cells/well) in triplicate and treated with PTX or nocodazole for 7 days. Colonies were fixed and stained with crystal violet, and well intensity or colony number was measured using the ImageJ software. All experiments report representative results of at least three independent repetitions.

**Immunofluorescence analysis**. U2OS and HCT116 cells were incubated in PBS-T buffer (0.2% Triton X-100 in 1 × PBS, phenylmethylsulphonylfluoride [PMSF], protease inhibitor cocktail [Sigma, P8340], and phosphatase inhibitor cocktail [Roche, P4906845001]) for 5 min on ice, followed by fixation with 2% paraformaldehyde. For kinetochore-localized BUB3 detection, cells were fixed with ice-cold 100% methanol for 15 min on ice. Primary antibody staining was performed as follows: anti-BUB3 (Abcam, ab133699, 1:500; Millipore, MABE1023, 1:200), anti-CUL4A (Abcam, ab92554, 1:100, Sigma, SAB1406671, 1:100), anti-DDB1 (Sigma, C9995, 1:100), anti-DDB1 (Cell Signaling, 5428, 1:100), anti-PML (Santa Cruz, sc-966, 1:500), anti-RepID (Bethyl Laboratories, A302-055A, 1:500), anti-CDT2 (Abcam, ab72264, 1:200), anti-VprBP (Abcam, ab202587, 1:80), anti-RBBP4 (Abcam, ab1765, 1:50), anti-RBBP7 (Abcam, ab3535, 1:50), anti-α-tubulin (Sigma, T9026, 1:500), and anti-CREST (ImmunoVision, HCT-0100, 1:1000) for 3 h at room temperature. Secondary antibody staining was performed as follows: Alexa 488- or 568-conjugated anti-mouse immunoglobulin G (IgG), IgG2b, Alexa 488- or 555-conjugated anti-rabbit IgG, and Alexa 647-conjugated anti-human IgG (1:500, Thermo Fisher Scientific, A11029, A21124, A21141, A11008, A21428, and A21445) for 1 h at room temperature.

A Zeiss LSM710 confocal microscope was used for imaging, and Pearson's correlation coefficient (R or Rcoloc)—which represents the covariance of the two

variables divided by the product of their standard deviations—was calculated using the colocalization Plugin of the FIJI-ImageJ software (https://imagej.nih.gov/ij/index.html) to carry out the colocalization analysis. For super-resolution microscopy, immunofluorescence slides were imaged with a VisiTech (Sunderland, UK) VT-iSIM super-resolution microscope, using a ×100 NA 1.45 Nikon PSF-optimized objective. Fluorophores were excited using the appropriate lasers, Diode 405 nm, Diode 488 nm, OPSL 561 nm, or Diode 642 nm, and super-resolution emission was collected with a 16-bit C-MOS camera (Hamamatsu, Japan). Z-sections were obtained at 100 nm intervals and deconvolved with the VisiTech proprietary software (Microvolution). Post-acquisition images were auto-adjusted for brightness and contrast using Image J (FIJI). Detailed results of the colocalization analyses are presented in Supplementary Dataset. For the time-lapse microscopic analysis, HCT116 WT and RepID KO cells stably expressing H2B-mTurquoise and geminin-mCherry were plated approximately 12 h prior to imaging. They were plated in full growth media on a collagen-coated glass-bottom 96-well dish (CellVis #P96-1.5H-N) so that the density would remain sub-confluent until the end of the imaging period. On the day of imaging, cells were treated with 1 µM CDK1 inhibitor (RO-3306) for 6 h to synchronize cells at the G2/M transition. Immediately prior to imaging, cells were washed with full growth media and placed on the microscope. Images were taken in CFP and RFP channels every 1 min on a Nikon Eclipse Ti2 microscope with a ×20 0.8 NA objective. Total light exposure time was kept to 200 ms for each time point. Cells were imaged in a humidified, 37 °C chamber in 5% $CO_2$. Image processing, cell tracking, and determination of geminin levels were carried out using custom Matlab scripts described previously[35]. The nuclear area was measured by determining the number of pixels in a nuclear mask generated from the H2B-mTurquoise signal. APC activity was determined as previously described[35].

**Chromatin fractionation, co-IP and immunoblotting**. Cells were harvested and incubated in cytosol extraction buffer containing NP-40 (20 mM Tris-HCl pH 7.4, 10 mM NaCl, 3 mM $MgCl_2$, 0.5% NP-40, PMSF, protease inhibitor cocktail, and phosphatase inhibitor cocktail). Cells were harvested by centrifugation at 2700 × g for 5 min at 4 °C, washed, and resuspended in nuclear extraction buffer (10 mM Tris-HCl pH 7.4, 100 mM NaCl, 1% Triton X-100, 1 mM EDTA pH 8, 1 mM EGTA, 0.1% sodium dodecyl sulfate (SDS), 10% glycerol, 0.5% sodium deoxycholate, protease inhibitor cocktail, and phosphatase inhibitor cocktail). The suspension was vortexed, incubated on ice, and then centrifuged at 5200 × g for 5 min at 4 °C. The pellet was resuspended with nuclear extraction buffer containing 5 mM $CaCl_2$ and micrococcal nuclease (New England Biolabs, Cat. M0247S), vortexed, and incubated at 37 °C for 5 min. Chromatin-bound fractions were collected after centrifugation at 18,000 × g for 5 min at 4 °C. Total cell lysates and chromatin-bound proteins were immunodetected following SDS-polyacrylamide gel electrophoresis (PAGE). Unless otherwise reported, all experiments exhibit representative results of at least three independent repetitions.

Soluble nuclear and/or chromatin-bound fractions were immunoprecipitated using an anti-FLAG (Sigma, F1804), anti-BUB3 (Abcam, ab133699), or anti-APC4 (Abcam, ab72149) at 4 µg of antibody per sample. After rotation overnight at 4 °C, 70 µl of Sepharose beads were added and samples were incubated for an additional 1 h at 4 °C with rotation. The protein–bead complexes were collected by centrifugation at 1700 × g for 3 min and washed three times with PBS. Seventy µl 2 × SDS sample loading dye were added, and the complexes were boiled for 10 min. Protein binding was immunodetected following SDS-PAGE. Unprocessed original scans of blots are shown in Supplementary Figs. 5 and 6.

The following primary antibodies were used: anti-RepID (NCI186, 1:1000), anti-PARP1 (Santa Cruz, sc-8007, 1:1000), anti-phosphorylated Ser10 histone H3 (Millipore, 06–570, 1:5000), anti-phosphorylated Ser28 histone H3 (Millipore, 07–145, 1:2000), anti-CUL1 (Abcam, ab75817, 1:1000), anti-CUL2 (Abcam, ab166917, 1:1000), anti-CUL3 (Abcam, ab75851, 1:20,000), anti-CUL4A (Abcam, ab92554, 1:20,000), anti-CUL4B (Sigma, C9995, 1:2000), anti-CUL5 (Abcam, ab184177, 1:5000), anti-CUL7 (Abcam, ab96861, 1:1000), anti-BUB3 (Abcam, ab133699, 1:10,000), anti-DDB1 (Cell Signaling, 5428, 1:1000), anti-MAD2 (Abcam, ab70383, 1:1000), anti-BUBR1 (Abcam, ab54894, 1:2000), anti-APC4 (Abcam, ab72149, 1:2000), anti-Cyclin B1 (Cell Signaling, 4138, 1:4000), anti-Securin (Abcam, ab79546, 1:10,000), anti-CDT2 (Abcam, ab72264, 1:1000), anti-SKP2 (Abcam, ab68455, 1:1000), anti-VprBP (Abcam, ab202587, 1:1000), anti-RBBP4 (Abcam, ab1765, 1:1000), anti-RBBP7 (Abcam, ab3535, 1:1000), anti-PML (Santa Cruz, sc-966, 1:1000), anti-FLAG (Sigma, F1804, 1:1000), anti-α-tubulin (Sigma, T9026, 1:2000), and anti-histone H3 (Millipore, 07–690, 1:20,000). For secondary antibodies, horseradish peroxidase (HRP)-linked anti-mouse IgG (Cell Signaling, 7076), HRP-linked anti-rabbit IgG (Cell Signaling, 7074) and HRP-linked anti-goat IgG (Santa Cruz, sc-2020) were used following the manufacturer's suggested protocols.

**In vivo ubiquitination assay**. HCT116 cells were transiently co-transfected with the indicated siRNA together with a HIS-tagged ubiquitin plasmid and incubated 48 h. Cells were treated with nocodazole first and, after 10 h, were treated with 10 µM MG132 (Calbiochem, 474790) for an additional 6 h. Cells were harvested at the indicated release points into MG132-containing fresh medium, lysed in denaturing buffer (6 M guanidine-HCl, 0.1 M $Na_2HPO_4/NaH_2PO_4$, and 10 mM imidazole), incubated with Ni-NTA agarose beads (QIAGEN, 1018244) for 3 h,

washed, boiled with Laemmli's buffer, and immunoblotted with anti-BUB3 antibody (Abcam, ab133699). All experiments report representative results of at least three independent repetitions.

**In vitro ubiquitination and binding assay**. For in vitro ubiquitination assay, FLAG-tagged plasmid of each CRL4 components (CUL4A, CUL4B, DDB1), DCAFs (RepID FL and its mutants or RBBP7), and BUB3 was transfected to HCT116 cells using Lipofectamine 2000 (Invitrogen) and purified using FLAG-Sepharose beads and FLAG-peptide (Sigma, A2220, F4799). Purified proteins were mixed with 1 μg ubiquitin (Sigma, U6253), 60 ng human recombinant E1, 300 ng UbcH5c, Mg-ATP solution, and ubiquitination buffer provided from Enzo Life Science (BML-UW9920-0001). After 60-min incubation at 30 °C, the reaction was denatured by adding SDS-containing loading buffer, boiled at 100 °C for 5 min, separated by SDS-PAGE, transferred to a PVDF membrane, and detected ubiquitinated BUB3 with anti-BUB3 antibody. For in vitro binding assay, purified proteins as indicated in the figure were mixed in lysis buffer and BUB3 was precipitated using anti-BUB3 antibody. Co-precipitated proteins were detected with anti-FLAG antibody. All experiments report representative results of at least three independent repetitions.

**Immunoprecipitation coupled to MS**. FLAG-RepID-overexpressed stable U2OS cells were lysed, and the mixture of soluble nuclear and chromatin-bound fractions were incubated with 4 μg of IgG as the negative control or anti-FLAG antibody (SIGMA, F1804) overnight at 4 °C with rotation. Seventy μl of Sepharose beads were added, and samples were incubated for an additional 1 h at 4 °C with rotation. The protein–bead complexes were collected by centrifugation at 1700 × g for 3 min and washed three times with PBS. Seventy μl 2 × SDS sample loading dye were added, and the complexes were boiled for 10 min. Coomassie blue-stained gel bands were cut into smaller pieces (10 separate bands per lane) and destained using 50% Acetonitrile with 25 mM ammonium bicarbonate, pH 8, with vortexing. Bands were then dried in a speed vacuum and rehydrated with 0.6 μg of trypsin in 25 mM ammonium bicarbonate, pH 8, (30 μl) on ice for 1 h. An additional 25 mM ammonium bicarbonate was then added to completely saturate the bands, and samples were incubated at 37 °C overnight. Peptides were extracted in 70% Acetonitrile and 5% formic acid using a bath sonication, and supernatant solutions were dried in the speed vacuum. Samples were desalted utilizing Pierce C18 spin columns (Thermo Fisher), dried, and resuspended in 0.1% trifluoroacetic acid prior to MS analysis. Peptides were analyzed on a Q Exactive (Thermo Scientific). The desalted tryptic peptide was loaded onto an Acclaim PepMap 100 C18 LC column (Thermo Scientific, CA) utilizing a Thermo Easy nLC 1000 LC system (Thermo Scientific, CA) connected to the Q Exactive mass spectrometer. Peptides were eluted with a 1–35% gradient of Acetonitrile with 0.1% formic acid over 75 min with a flow rate of 300 nl/min. The QE was operated with each MS1 scan in the orbitrap at 70,000 resolution with a maximum injection time of 256 ms and an AGC target of 1e6. The MS2 scans had a normalized collision energy of 25 and were run at 17,500 resolution with a maximum injection time of 64 ms and an AGC target of 1e5.

The raw MS data were collected and analyzed in Proteome Discoverer 2.1 (Thermo Scientific) with the Sequest HT software and was searched against the Human Proteome database. The parent ion mass tolerance was set to 10 ppm and the fragment ion mass was 0.6 Da. Trypsin was set as the digestion enzyme and the minimal peptide length was six amino acids with a maximum of two missed cleavages allowed.

## Data availability

The source data underlying Figs. 1b, c, 2a, 4d, 5a–f, h, and 6f and Supplementary Figs. 1a, b, 2a, d, e, 3h, and 4a, c, g are provided as a Source Data file. All data within the manuscript are available from the authors upon request.

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

## Acknowledgements

We are grateful to Dr. Michael Kruhlak, Dr. Langston Lim, and Dr. Andy Tran (Confocal Microscopy Core Facility, CCR, NCI, NIH) for expert technical assistance in confocal microscopy; Dr. Ukhyun Jo for the HIS-tagged ubiquitin plasmid. We thank Dr. Mary Dasso, Dr. Munira Basrai, and Dr. Robin Sebastian for helpful discussions. This study was supported by the Intramural Research Program of the NIH, Center for Cancer Research, National Cancer Institute and the IRP program, National Institute on Aging.

## Author contributions

S.M.-J. and M.I.A. designed the study. S.M.-J., J.F.N., F.E.I., S.D.C. and M.I.A. designed the experiments. S.M.-J., J.F.N., L.M.J., M.J.O., F.E.I. and S.D.C. performed the experiments. S.M.-J., J.F.N., H.F., C.E.R., L.M.J., L.S.P., J.M.G., M.J.O., F.E.I., S.D.C. and M.I.A. analyzed the data. S.M.-J., J.F.N., H.F., C.E.R., B.L.T., L.S.P., A.M.B., J.M.G., F.E.I., S.D.C. and M.I.A. wrote the manuscript.

## Competing interests

The authors declare no competing interests.
