## [Peer Review File · Nature Communications]

Reviewers' comments:

Reviewer #1 (Remarks to the Author):

Comments on

Jang et al

The RepID-CRL4 ubiquitin ligase complex regulates metaphase to anaphase transition via BUB3 degradation

In this manuscript, Jang et al first made an observation that RepID depleted cells exhibited delayed mitotic exit and then followed up with mechanistic studies. RepID is an adaptor protein that recruits Cullin-RING ubiquitin ligase complex 4 (CRL4) to chromatin prior to DNA replication. Their initial observation suggested that RepID/CRL4 also plays a role in mitosis progression. The authors found that RepID interacts with BUB3, a spindle assembly checkpoint protein, particularly during G2/M phase. The authors showed that BUB3 protein level dropped as HCT116 cells exit from mitosis, due to CRL4-mediated ubiquitylation. RepID is required for maintaining CRL4 at condensed chromosomes during mitosis but does not affect BUB3 localization. Surprisingly, RepID was found to dissociate from CRL4 before BUB3 degradation, suggesting another substrate-binding adaptor protein is responsible for BUB3 degradation. The authors identified RBBP7 as the adaptor protein, and proposed CRL4 switched adaptor proteins at the metaphase-anaphase transition and CRL4-RBBP7 mediates BUB3 degradation for mitotic exit. According to the authors, interphase BUB3 was spared of degradation partially due to sequestration in the PML nuclear bodies. They also showed that RepID-CRL4-RBBP7 depletion sensitized cells to microtubule stabilizing drug paclitaxel.

The authors supported each individual conclusion with multiple lines of evidence, and the quality of data was generally high. A relatively complete story could be seen from several key pieces of discoveries presented here. However, obvious gaps exist for the story, for which at least some discussions are warranted. It will also be helpful if they integrate the prior information of different ubiquitin ligases in mitosis progression into their discussions. Additionally, some data deviations have to be addressed.

Major points:

1. Which CRL4 complex degrades BUB3 and when where?

-Fig 7 suggests that during interphase CRL4-Cdt2 ubiquitylates BUB3 for degradation sparing only the BUB3 sequestered by PML nuclear bodies. Cdt2 only associates with S phase chromatin. Based on Fig 3C, BUB3 level seems to go up during G2/M phase. Is there a connection here? Some experiments can be done to solidify or reject the connection.

-Fig 7 and Fig 4g summarizes the CRL4-RepID switch to CRL4-RBBP7 for BUB3 degradation. If RepID is responsible for CRL4 chromatin recruitment, after its dissociation what holds CRL4 at kinetochores/chromatin? or is the localization necessary for its ubiquitylation of BUB3?

Immunofluorescence (IF) of CUL4A/B/DDB1 in mitotic cells (similar to Fig 4C) will be informative.

-Fig4c and Fig 4d, you cannot claim RBBP7 and BUB3 co-localize based on Fig 4d. RBBP7 staining looks like the spindle in Fig 4d, but more kinetochore-like in Sup Fig 3b? This needs to be clarified: Sup Fig 3B needs co-staining with BUB3, and both Sup Fig 3B and Fig 4d needs zoom-in to show details at kinetochores.

-Compare Fig 3e and Fig 4f. BUB3 ubiquitylation is abolished in the absence of either RepID or RBBP7. How could RepID knockout affect BUB3 ubiquitylation if RepID does not have catalytic role? Need show whether RBBP7 and CUL4 co-IP in RepID deficient nocodazole released cells. This is the KEY missing evidence to support the "adaptor handover" model shown in Fig 4G and Fig 7.

2. Effects of CRL4-RepID-RBBP on mitosis progression

-Fig 1d. RepID knockout cells shows about 20 min delay in anaphase onset, why does the FACS in Fig 1b show ~6hr delay?

-Fig 1 shows RepID KO cells released from nocodazole accumulated subG1 apoptotic cells.

However, "growth assays showed that RepID WT and KO cells exhibit similar sensitivities to the

microtubule polymerization inhibitor nocodazole (Fig. 5d)". Why?

3. It is worth mentioning that CUL4-RBBP7 was indicated in CENP-A loading after mitotic exit (e.g. Mousset J et al., JCS, 2015). Whether this is related to BUB3 ubiquitination described here might be interesting to compare. Other late mitotic events regulated by non-APC/C mediated ubiquitylation: CUL3 complexes are required for aurora B localization (Sumara, I. et al, Dev Cell, 2007).

Minor points:

1. BUB3 is often not regarded as a key player for physical inhibition of the APC/C. What do the authors think about that?
2. BUB3 level is constant throughout the cell cycle in HeLa cells. One major difference is that the HCT116 cells used in this study have functional p53. Does p53 status affect BUB3 ubiquitination?
3. Fig 2c d: it seemed BUB3 has some interaction with CUL2.
4. Fig 3A: no change of kinetochore BUB3 level in RepID knockout cells. Is it because of the IF protocol or because the cells are in prometaphase?
5. Are IFs in fig 6b,d,e done following exactly the same protocol? The differences in the level and localization of BUB3 are so striking.

Reviewer #2 (Remarks to the Author):

In this manuscript, the authors identified a novel role of RepID-CRL4, known for modulating DNA replication, in the regulation of the degradation of BUB3, which triggers the termination of SAC and enables chromosome segregation. They investigated the mechanism and found that in mitosis, RepID is disassociated from CRL4 and replaced by RBBP7. RBBP7 ubiquitinates BUB3 and triggers BUB3 degradation, leading to mitotic exit. This study identified a previously unrecognized role of RepID in mitotic checkpoint and demonstrated an interesting switch of DCAFs to regulate the progression of cell cycle. The manuscript is well written and the data are clearly presented. However, several points still need to be addressed.

1. The mitotic phenotype of RepID KO cells is interesting, but it is important to confirm the observation by showing that the mitotic defects can be corrected by reconstitution of RepID. In addition to HCT116, KO or knock-down of RepID in other cell lines is needed to confirm that this is a general phenomenon.
2. For the interaction of RepID with BUB3 shown in Fig.2B, is the WD40 domain of RepID sufficient to mediate the interaction with BUB3? Does BUB3 use the same domain for the interactions with RepID and RBBP7? This could be a possible mechanism for DCAF switching. The mechanism of how switching of RepID to RBBP7 is induced in mitosis is not addressed in this report.
3. Ubiquitination of BUB3 depends on both RepID and RBBP7 in vivo. In vitro ubiquitination assay would be needed to show that BUB3 is truly the substrate of RBBP7-containing CRL4.
4. In Fig4b, the interaction of RBBP7 with CUL4 but not with BUB3 is shown. Is the interaction between RBBP7 and BUB3 increased in mitosis? In the absence of RepID, is the interaction between RBBP7 and BUB3 affected when cells enter the mitosis?

Reviewer #3 (Remarks to the Author):

In this manuscript Aladjem and colleagues investigate BUB3 degradation during the cell cycle. BUB3 is a critical component of the spindle assembly checkpoint (SAC). Degradation of BUB3 would silence the SAC and promote anaphase onset. The authors investigate the role of the CRL4 E3 ubiquitin ligase in BUB3 degradation. Their major conclusions are that CRL4 is recruited to chromatin by RepID, that during mitosis CRL4 dissociates from RepID and binds to RBBP7 which is

the complex of CRL4 that ubiquitinates BUB3. BUB3 is protected from CRL4-mediated ubiquitination during interphase by association with PML. The findings that RepID deficient cells were delayed exiting mitosis and entering G1, and showed compromised geminin, cyclin B and securin degradation, is the starting point for this study with the proposal that RepID 'passes over' CRL4 to RBBP7 in mitosis. This is linked to increased levels of BubR1 associated with the APC/C in RepID deficient cells. Bub3 levels decline in RepID proficient but not RepID deficient cells after release from exposure to nocodazole. This is UPS-dependent. CUL4A-B and RBBP7 depletion also leads to reduced BUB3 degradation.

Altogether the authors perform a wide range of cell-based experiments. However overall, their model is not completely convincing without further experiments to validate the proposal that CRL4-RBBP7 is the E3 ligase directly responsible for ubiquitinating BUB3.

Questions and comments.

1. The authors find that Bub3 dissociates from RepID before Bub3 degradation. Did the authors test the timing of Bub3 ubiquitination relative to Bub3 degradation?
2. It wasn't clear what is the evidence that CRL4 transitions from CRL4-RepID to CRL4-RBBP7, and what is the mechanism.
3. In Fig. 4e the authors show that in RBBP7 deficient cells there is a delayed mitotic exit, but the effect for CUL4 depletion appears quite small. Is there an additive effect with RepID deficient cells? One would presume not if RBBP7 is downstream of RepID.
4. The authors do not provide evidence that either RBBP7 and/or CRL4-RBBP7 interacts with BUB3. The authors should show interaction between BUB3 and CRL4-RBBP7 through a co-IP, and a direct interaction in vitro with a reconstituted system.
5. Although the authors show that RBBP7 depletion in cells reduces BUB3 ubiquitination (Fig. 4f), this is not evidence that CRL4-RBBP7 itself ubiquitinates BUB3. The same results were observed for RepID (Fig. 3e) but the authors are not concluding that RepID ubiquitinates BUB3.
6. The major concern is the absence of evidence that CRL4-RBBP7 ubiquitinates BUB3 in vitro. For this model to be convincing the authors should show that reconstituted CRL4-RBBP7 ubiquitinates BUB3.
7. In addition since the authors claim that CRL4-RepID does not ubiquitinate BUB3, even though RepID interacts with BUB3 (whether this is direct or indirect was not shown), the activity of CRL4-RepID towards BUB3 should also be tested.
8. The authors appear to assume that is kinetochore-associated BUB3 that is degraded by CRL4-RBBP7. However have they considered BUB3 in the complex with the mitotic checkpoint complex?
9. Lines 67-69. The mechanism of SAC silencing is incorrect. The APC/C is activated because the MCC dissociates from the APC/C-Cdc20 complex due to Cdc20 (of the MCC) of BubR1 ubiquitination.

Point by point response to reviewers' comments:

We thank all the reviewers for their thoughtful evaluation of our original submission and for their helpful suggestions. We have revised the manuscript based on the reviewers' comments. We believe that reviewers' suggestions have significantly improved the paper, and we appreciate the reviewers' time and help. Below is our detailed response to the reviewers' comments.

Reviewer #1 (Remarks to the Author):

In this manuscript, Jang et al first made an observation that RepID depleted cells exhibited delayed mitotic exit and then followed up with mechanistic studies. RepID is an adaptor protein that recruits Cullin-RING ubiquitin ligase complex 4 (CRL4) to chromatin prior to DNA replication. Their initial observation suggested that RepID/CRL4 also plays a role in mitosis progression. The authors found that RepID interacts with BUB3, a spindle assembly checkpoint protein, particularly during G2/M phase. The authors showed that BUB3 protein level dropped as HCT116 cells exit from mitosis, due to CRL4-mediated ubiquitylation. RepID is required for maintaining CRL4 at condensed chromosomes during mitosis but does not affect BUB3 localization. Surprisingly, RepID was found to dissociate from CRL4 before BUB3 degradation, suggesting another substrate-binding adaptor protein is responsible for BUB3 degradation. The authors identified RBBP7 as the adaptor protein, and proposed CRL4 switched adaptor proteins at the metaphase-anaphase transition and CRL4-RBBP7 mediates BUB3 degradation for mitotic exit. According to the authors, interphase BUB3 was spared of degradation partially due to sequestration in the PML nuclear bodies. They also showed that RepID-CRL4-RBBP7 depletion sensitized cells to microtubule stabilizing drug paclitaxel.

The authors supported each individual conclusion with multiple lines of evidence, and the quality of data was generally high. A relatively complete story could be seen from several key pieces of discoveries presented here. However, obvious gaps exist for the story, for which at least some discussions are warranted. It will also be helpful if they integrate the prior information of different ubiquitin ligases in mitosis progression into their discussions. Additionally, some data deviations have to be addressed.

Response: We thank the reviewer for this evaluation of the manuscript and its significance, and for the assessment of the quality of the data. We have addressed all the gaps identified by the reviewer, either by including additional data or by expanding the discussion as suggested. Detailed responses to all the points raised by the reviewer are presented below.

Comment: 1. Which CRL4 complex degrades BUB3 and when where?

-Fig 7 suggests that during interphase CRL4-Cdt2 ubiquitylates BUB3 for degradation sparing only the BUB3 sequestered by PML nuclear bodies. Cdt2 only associates with S phase chromatin. Based on Fig 3C, BUB3 level seems to go up during G2/M phase. Is there a connection here? Some experiments can be done to solidify or reject the connection.

HCT116 RepID WT cells transfected with FLAG-tagged BUB3 and treated with siRNA-CTL or siRNA-PML were synchronized by double thymidine and released into fresh medium. Chromatin fractions were immunoprecipitated by FLAG antibody (detecting BUB3) and analyzed by immunoblotting with the indicated antibodies.

Response: We thank the reviewer for raising this point and for suggesting this connection between BUB3 and CDT2 levels. Our data suggest that BUB3 can be degraded by two CRL4 complexes: in mitosis, CRL4^{RBBP7} degrades BUB3; during interphase, CRL4^{CDT2} can potentially degrade BUB3, but this degradation is inhibited if BUB3 associates with PML bodies. To test the interaction between BUB3 and CDT2 directly, we have performed co-IP experiments now presented in Supplementary Fig. 4i (also shown here on the left). These experiments indeed show that the interaction between CDT2

and BUB3 is S-phase specific and that it occurs only in PML-depleted cells. As the reviewer has suggested, beyond corroborating the hypothesis that the CDT2-BUB3 interaction is prevented by PML, BUB3 accumulation in G2 might, therefore, also reflect the absence of CDT2-mediated degradation of BUB3. Eventually, the accumulation of undegraded BUB3 molecules builds a pool of BUB3 molecules that can potentially mediate the SAC during mitosis. This point is discussed in the revision (page 14, last paragraph).

Comment 1 (cont): -Fig 7 and Fig 4g summarizes the CRL4-RepID switch to CRL4-RBBP7 for BUB3 degradation. If RepID is responsible for CRL4 chromatin recruitment, after its dissociation what holds CRL4 at kinetochores/chromatin? or is the localization necessary for its ubiquitinylation of BUB3? Immunofluorescence (IF) of CUL4A/B/DDB1 in mitotic cells (similar to Fig 4C) will be informative.

Immunofluorescence analysis using RBBP7, CUL4A and CREST antibodies with DAPI staining in HCT116 RepID WT and KO cells.

Response: Thank you for this suggestion, we agree that it is important to characterize the association of CUL4 with chromatin directly. In the revised version, we have analyzed the localization of CUL4A during mitosis as suggested using immunofluorescence and super resolution microscopy (new Supplementary Fig. 3f, also shown to the left). These images confirm that CUL4A localizes to mitotic chromosomes only in RepID proficient cells. We agree with the reviewer’s interpretation that its recruitment, but not retention, on metaphase chromosomes requires RepID. We conclude that the association of CUL4 with chromatin (as a consequence of recruitment by RepID) is required for degradation of BUB3, although the catalytic DCAF mediating this degradation is RBBP7. This specific point is discussed in the revision (page 13, last paragraph).

Comment 1 (cont): -Fig4c and Fig 4d, you cannot claim RBBP7 and BUB3 co-localize based on Fig 4d. RBBP7 staining looks like the spindle in Fig 4d, but more kinetochore-like in Sup Fig 3b? This needs to be clarified: Sup Fig 3B needs co-staining with BUB3, and both Sup Fig 3B and Fig 4d needs zoom-in to show details at kinetochores.

Response: Thank you, we agree that more information about the localization of RBBP7 and BUB3 is needed. In the revision we have replaced Fig. 4d with a new imaging study in which we performed triple staining for RBBP7, BUB3, and CREST during metaphase (new Fig. 4d), and added Supplemental Fig. 3e with images of RBBP7 and tubulin to test for spindle-like localization of RBBP7 (both images also pasted below). This co-staining study suggests that RBBP7 colocalizes with the mitotic spindle and associates with kinetochores. Consistent with the role of CRL4^{RBBP7} in the degradation of BUB3, the association between RBBP7 and BUB3 during metaphase was stronger in RepID-deficient cells, which do not recruit CRL4 to mitotic chromatin (see above), than in RepID-proficient cells, in which CRL4 is recruited to chromatin and facilitates BUB3 degradation. To further probe into these interactions, we also tested for RBBP7, BUB3, and CREST localization in mitotic cells after exposure to a p97 inhibitor, which prevented the degradation of BUB3, and observed an increased association between BUB3 and RBBP7. This observation is similar to what we observed in the absence of RepID, when CRL4 was depleted and BUB3 degradation was inhibited.

A super-resolution microscopy analysis using RBBP7, BUB3 and CREST antibodies with DAPI staining in RepID WT and KO cells with and without CB5083, a p97 inhibitor.

A super-resolution microscopy analysis using anti-RBBP7 and tubulin antibodies with DAPI staining in RepID WT and KO cells.

Comment 1 (cont): -Compare Fig 3e and Fig 4f. BUB3 ubiquitylation is abolished in the absence of either RepID or RBBP7. How could RepID knockout affect BUB3 ubiquitylation if RepID does not have catalytic role? Need show whether RBBP7 and CUL4 co-IP in RepID deficient nocodazole released cells. This is the KEY missing evidence to support the “adaptor handover” model shown in Fig 4G and Fig 7.

Response: Yes, thank you, we agree that this is a key point and it is important to emphasize it. Our hypothesis, based on our observation that RepID is required to recruit CRL4 is recruited to chromatin (PMID: 30018425, Ref #14), suggests that RepID-deficient cells show lower levels of BUB3 ubiquitylation because of CRL4’s absence from chromatin in those cells. As the reviewer has suggested, we tested this hypothesis directly in the revision. The revised submission includes new co-immunoprecipitation data measuring interactions between RBBP7 and CRL4 in chromatin fractions following release from nocodazole (new Supplementary Fig. 3b, also shown below). As expected, CRL4 chromatin-bound levels (input) were lower in RepID-KO cells, but during the first 30 min after release from a nocodazole block, both KO and WT cells exhibited a similar fraction of RBBP7-bound BUB3 and CRL4 (CUL4A/DDB1). We have also tested BUB3 ubiquitination with purified proteins to assess the catalytic role of both RBBP7 and RepID (Supplemental Fig. 3i, below). The results suggest that RBBP7 is the catalytic DCAF, and the absence of BUB3

ubiquitylation in RepID deficient cells reflects the fact that CRL4 is not recruited to chromatin in those cells. RBBP7 binds CRL4 after RepID's dissociation, and plays a catalytic role in BUB3 ubiquitylation, supporting the adaptor handover model.

b

i

Supplementary Fig. 3: (b) HCT116 RepID WT and KO cells transfected with FLAG-tagged RBBP7 were synchronized by nocodazole and released into fresh medium. Chromatin fractions were immunoprecipitated by FLAG antibody (detecting RBBP7) and analyzed by immunoblotting with the indicated antibodies. Binding ratio was calculated by dividing input, followed by precipitated RBBP7 level. (i) In vitro ubiquitination assay using purified proteins, followed by immunoblotting with BUB3 antibody.

Comment: 2. Effects of CRL4-RepID-RBBP on mitosis progression

-Fig 1d. RepID knockout cells shows about 20 min delay in anaphase onset, why does the FACS in Fig 1b show ~6hr delay?

Response: Thank you for pointing out this apparent discrepancy, we agree that this point should indeed be discussed. First, the discrepancy observed between the two modes of cell synchronization can be explained by the fact that the use of nocodazole can introduce delay in G2/M release. The live imaging analysis presented in Fig. 1d is following cells' release from CDK1 inhibitor while the

HCT116 RepID WT and KO cells were released in fresh media from the mitotic block and collected every 30 minutes, followed by flow cytometry analysis.

FACS in Fig. 1b was obtained after using nocodazole, which is known to activate a pathway that delays the G2/M transition, whereby microtubules are disassembled and chromosomes are transiently decondensed (PMID: 10996076, reference #37 in the revision). In addition, nocodazole can generate DNA damage that could result in mitotic delay (PMID: 20660628, reference #38 in the revision). These effects of nocodazole are mentioned in the revision (page 5, starting at the last sentence). In order to document the delay after nocodazole release, we have also included new data in the revised manuscript showing short-interval FACS analysis (new Supplementary Fig. 1a, also shown to the left).

Comment 2 (cont): -Fig 1 shows RepID KO cells released from nocodazole accumulated subG1 apoptotic cells. However, “growth assays showed that RepID WT and KO cells exhibit similar sensitivities to the microtubule polymerization inhibitor nocodazole (Fig. 5d)”. Why?

Response: Again we thank the reviewer for pointing this out, as we need to clarify that the extent of acute and chronic responses to nocodazole cannot be compared directly. Because acute exposure to nocodazole induces prolonged mitotic arrest that can induce apoptosis, we were avoiding a highly toxic dose for the colony formation assay and used a lower concentration of nocodazole for this assay (less than 30 nM vs. 100 nM when employing nocodazole to induce a complete cell cycle blockage). The acute apoptosis we have observed in RepID deficient cells using a high dose of nocodazole did not translate to a significant loss of viability in long-term assays using a lower dose. This issue is discussed in the revised paper (page 10, end of the first paragraph).

Comment: 3. It is worth mentioning that CUL4-RBBP7 was indicated in CENP-A loading after mitotic exit (e.g. Mouysset J et al., JCS, 2015). Whether this is related to BUB3 ubiquitination described here might be interesting to compare. Other late mitotic events regulated by non-APC/C mediated ubiquitylation: CUL3 complexes are required for aurora B localization (Sumara, I. et al, Dev Cell, 2007).

Response: We thank the reviewer for this suggestion. Based on the reviewer’s suggestion, we have discussed these important observations in the revision (references #40 and #56).

Minor points:

Comment:

1. BUB3 is often not regarded as a key player for physical inhibition of the APC/C. What do the authors think about that?

Response: Thank you for this important comment, and indeed, we wish to highlight that our study emphasizes that BUB3 plays a more critical role in mitosis than originally thought. Our understanding is that the activity of the APC/C is regulated throughout the cell cycle by several mechanisms, one involving BUB3, which would interact with CDC20 and inhibit the formation of APC/C-CDC20. Our data are in line with observations in mice carrying a disruption of the *Bub3* gene, which leads to embryonic lethality of lagging chromosomes, micronuclei and chromatin bridging among others (PMID: 10995385, #63 in the revision), and in observations suggesting that patients carrying the *BUB3* mutations show mosaic aneuploidy (PMID: 23747338; #64 in the revision). These observations are discussed in the revised submitted paper (page 15, first paragraph).

Comment: 2. BUB3 level is constant throughout the cell cycle in HeLa cells. One major difference is that the HCT116 cells used in this study have functional p53. Does p53 status affect BUB3 ubiquitination?

Response: We thank the reviewer for this very interesting suggestion. We tested the hypothesis that p53 status affects BUB3 degradation directly by generating RepID KO in two additional cell lines with dysfunctional p53: H1299 (a Non-Small Cell Lung Cancer, p53^{-/-}) and DMS114 (Small Cell Lung Cancer, mutant p53). The new data show (new Supplementary Fig. 2b-f, also shown below) that all RepID KO cell lines exhibit a mitotic delay after nocodazole release and suggest that the delayed mitotic and BUB3 ubiquitination are independent of the status of p53.

Supplementary Fig. 2: (d) RepID WT, KO and reconstituted RepID FL in KO background in three cell lines were synchronized by nocodazole, released in fresh media and collected after 3 hours, followed by flow cytometry. (e) Percentage of cells in G1 and G2/M.

Comment: 3. Fig 2c d: it seemed BUB3 has some interaction with CUL2.

Response: Yes, thanks, as noticed by the reviewer, there may be a marginal interaction between BUB3 and CUL2. This interaction is much lower than the observed interaction between BUB3 and CUL4, and is not evident in cell cycle fractionated cells (Fig. 2d).

Comment: 4. Fig 3A: no change of kinetochore BUB3 level in RepID knockout cells. Is it because of the IF protocol or because the cells are in prometaphase?

Response: Thank you, we agree, the reviewer is correct. There was no change of kinetochore BUB3 levels in RepID-proficient and deficient cells because the cells were in prometaphase.

Comment: 5. Are Ifs in fig 6b,d,e done following exactly the same protocol? The differences in the level and localization of BUB3 are so striking.

Response:

Yes, the IFs in Fig. 6b, d, e were performed using a same protocol and with identical thresholds during image captures to prevent signal saturation. To present a better indication of signal distribution, in the revised submission we provided additional images containing a larger number of cells per field (new Supplementary Fig. 4d-f. left).

Detection of BUB3 in RepID WT or KO cells with/without siRNA-PML or MG132.

Reviewer #2 (Remarks to the Author):

Comment: In this manuscript, the authors identified a novel role of RepID-CRL4, known for modulating DNA replication, in the regulation of the degradation of BUB3, which triggers the termination of SAC and enables chromosome segregation. They investigated the mechanism and found that in mitosis, RepID is disassociated from CRL4 and replaced by RBBP7. RBBP7 ubiquitinates BUB3 and triggers BUB3 degradation, leading to mitotic exit. This study identified a previously unrecognized role of RepID in mitotic checkpoint and demonstrated an interesting switch of DCAFs to regulate the progression of cell

Cell cycle analysis after release from mitotic block (d-e), and in vivo ubiquitination analysis (f) in HCT116, H1299 and DMS114 cells with WT, KO and reconstituted RepID.

recovered mitotic exit and BUB3 ubiquitination in three cell lines, suggesting that the mitotic problems observed in RepID-deficient cells are caused by RepID deficiency. For these analyses, we have generated RepID KO in two additional cell lines: H1299 (Non-Small Cell Lung Cancer) and DMS114 (Small Cell Lung Cancer), both showing a mitotic delay that can be prevented by reconstitution with RepID.

Comment: 2. For the interaction of RepID with BUB3 shown in Fig.2B, is the WD40 domain of RepID sufficient to mediate the interaction with BUB3? Does BUB3 use the same domain for the interactions with RepID and RBBP7? This could be a possible mechanism for DCAF switching. The mechanism of how switching of RepID to RBBP7 is induced in mitosis is not addressed in this report.

Response: We thank the reviewer for this comment, which raises an important and relevant point. The WD domain by itself is not sufficient to mediate interaction, but we performed additional experiments using a RepID construct lacking the RepID WD40 domain (F2-5 construct). Our results suggest that depletion of the WD40 domain of RepID was sufficient to prevent the interaction with BUB3. We added the new data in the revised Fig. 2b (right).

cycle. The manuscript is well written, and the data are clearly presented. However, several points still need to be addressed.

Response: We thank the reviewer for this evaluation of the manuscript, as well as for the important comments and suggestions. Below we address the specific points raised by the reviewer.

Comment: 1. The mitotic phenotype of RepID KO cells is interesting, but it is important to confirm the observation by showing that the mitotic defects can be corrected by reconstitution of RepID. In addition to HCT116, KO or knock-down of RepID in other cell lines is needed to confirm that this is a general phenomenon.

Response: We thank the reviewer this suggestion, we agree that reconstitution experiments are critical. We have included experiments with reconstituted RepID in RepID-deficient cells in the revised submission (Supplementary Fig. 2d-f, shown to the left). These experiments demonstrate that reconstituted RepID (FL)

Soluble nuclear and chromatin-bound fractions from U2OS cells expressing the indicated FLAG-RepID mutants were immunoprecipitated with FLAG antibodies and analyzed by immunoblotting.

Comment: 3. Ubiquitination of BUB3 depends on both RepID and RBBP7 in vivo. In vitro ubiquitination assay would be needed to show that BUB3 is truly the substrate of RBBP7-containing CRL4.

Response:

We thank to reviewer for this suggestion. As suggested, we performed in vitro ubiquitination assays with immunopurified flagged-CUL4A, CUL4B, DDB1, RepID, BUB3 and RBBP7. The revised manuscript reports these assays and shows that RBBP7 (but not RepID) is a catalytic DCAF required to ubiquitinate BUB3 (new Supplementary Fig. 3i, left).

Comment: 4. In Fig4b, the interaction of RBBP7 with CUL4 but not with BUB3 is shown. Is the interaction between RBBP7 and BUB3 increased in mitosis? In the absence of RepID, is the interaction between RBBP7 and BUB3 affected when cells enter the mitosis?

Response: We thank the reviewer for this comment, which raises an interesting point. In the revision, we include experiments examining the interactions between RBBP7 and CRL4 and between RBBP7 and BUB3. Interactions were quantified in nocodazole-released HCT116 WT and RepID KO cells. The data are presented in the new Supplementary Fig. 3b (right). Indeed, interactions between RBBP7 and CUL4A or between RBBP7 and BUB3 increased in metaphase (30 min post release; at later time points, BUB3 is degraded in RepID WT cells so interactions cannot be detected).

Reviewer #3 (Remarks to the Author):

Comment: In this manuscript Aladjem and colleagues investigate BUB3 degradation during the cell cycle. BUB3 is a critical component of the spindle assembly checkpoint (SAC). Degradation of BUB3 would silence the SAC and promote anaphase onset. The authors investigate the role of the CRL4 E3 ubiquitin ligase in BUB3 degradation. Their major conclusions are that CRL4 is recruited to chromatin by RepID, that during mitosis CRL4 dissociates from RepID and binds to RBBP7 which is the complex of CRL4 that ubiquitinates BUB3. BUB3 is protected from CRL4-mediated ubiquitination during interphase by association with PML. The findings that RepID deficient cells were delayed exiting mitosis and entering G1, and showed compromised geminin, cyclin B and securin degradation, is the starting point for this study with the proposal that RepID 'passes over' CRL4 to RBBP7 in mitosis. This is linked to increased levels of BubR1 associated with the APC/C in RepID deficient cells. Bub3 levels decline in RepID proficient but not RepID deficient cells after release from exposure to nocodazole. This is UPS-dependent. CUL4A-B and RBBP7 depletion also leads to reduced BUB3 degradation.

Altogether the authors perform a wide range of cell-based experiments. However overall, their model is not completely convincing without further experiments to validate the proposal that CRL4-RBBP7 is the E3 ligase directly responsible for ubiquitinating BUB3.

Response: We thank the reviewer for the evaluation of the manuscript and for the comments and suggestions. We have addressed the specific points raised by the reviewer as described below.

Comment: 1. The authors find that Bub3 dissociates from RepID before Bub3 degradation. Did the authors test the timing of Bub3 ubiquitination relative to Bub3 degradation?

Response: We thank the reviewer for rising this point. Our results show that RepID dissociated from CRL4 prior to BUB3 degradation; RBBP7 was incorporated into CRL4 concomitant with RepID dissociation (Fig. 4a,b, Supplementary Fig. 3b, below), and BUB3 ubiquitination was concomitant with both its reduced abundance (Fig. 3e, 3f, 4f) and APC/C activation (Supplementary Fig. 1e,f) – all occurring within 60 minutes after nocodazole release.

Fig. 4: HCT116 cells with the indicated FLAG-tagged constructs were released from a nocodazole block and proteins were isolated at the indicated time. Chromatin and whole cell fractions were immunoprecipitated by the FLAG antibody and analyzed with immunoblotting with the indicated antibodies.

Supplementary Fig. 3b: HCT116 RepID WT and KO cells transfected with FLAG-tagged RBBP7 were synchronized by nocodazole and released into fresh medium. Chromatin fractions were immunoprecipitated by FLAG antibody (detecting RBBP7) and analyzed by immunoblotting with the indicated antibodies.

Comment: 2. It wasn't clear what is the evidence that CRL4 transitions from CRL4-RepID to CRL4-RBBP7, and what is the mechanism.

Response: We thank the reviewer for this comment, which was also raised by the other reviewer. To address this question, we performed a series of reciprocal co-immunoprecipitation experiments (Fig. 4a,b and Supplementary Fig. 3b, above). These studies show that RepID dissociates from chromatin-bound CRL4 shortly after the onset of metaphase (15-20 minutes after nocodazole release) as is evident by loss of co-precipitation of CUL4A, CUL4B and DDB1 with FLAG-RepID and a reciprocal loss of co-precipitation of RepID with FLAG-DDB1. Concomitantly, RBBP7 is recruited to chromatin and interacts with DDB1 (co-precipitation of RBBP7 with FLAG-DDB1 and reciprocal co-precipitation of CUL4A and DDB1 with FLAG-RBBP7). We also observed (Supplementary Fig. 3a) that RBBP7 does not interact directly with RepID. These results suggest that RepID mediated recruitment of CRL4 to chromatin is critical for the activity of RBBP7 on BUB3; in RepID WT cells, the interaction of CUL4 with RepID precedes the incorporation of RBBP7 into chromatin-bound CRL4 and facilitates RBBP7-mediated BUB3 ubiquitination.

Comment: 3. In Fig. 4e the authors show that in RBBP7 deficient cells there is a delayed mitotic exit, but the effect for CUL4 depletion appears quite small. Is there an additive effect with RepID deficient cells? One would presume not if RBBP7 is downstream of RepID.

HCT116 RepID WT and KO cells transfected with indicated siRNA were synchronized by nocodazole, released into fresh medium, and cell cycle stages

Response: We thank the reviewer for this interesting and helpful suggestion. As suggested, we performed new experiments to examine if the effects of RBBP7 and RepID were additive by examining how depletion of RBBP7 and/or CUL4 in WT and RepID KO cells would affect release from nocodazole. Data were compiled in the new Supplementary Fig. 3h of the revised manuscript (see figure) and demonstrate that RBBP7 was indeed downstream of RepID.

Comment: 4. The authors do not provide evidence that either RBBP7 and/or CUL4-RBBP7 interacts with BUB3. The authors should show interaction between BUB3 and CUL4-RBBP7 through a co-IP, and a direct interaction in vitro with a reconstituted system.

Response: We thank to reviewer for this suggestion, yes, we agree that this point needed to be clarified. In the revision, we performed *in vitro* ubiquitination and protein interaction assays with immunopurified CUL4A, CUL4B, DDB1, RepID, BUB3 and RBBP7. The revised submission includes data that show a direct interaction between BUB3 and CUL4/RBBP7 as well as *in vitro* binding assays with a reconstituted system using purified proteins (new Supplementary Fig. 3b and c, also shown below).

b

c

Supplementary Fig. 3: (b) HCT116 RepID WT and KO cells transfected with FLAG-tagged RBBP7 were synchronized by nocodazole and released into fresh medium. Chromatin fractions were immunoprecipitated by FLAG antibody (detecting RBBP7). (c) *In vitro* binding assay was performed by pull-down of BUB3 using its antibody in mixture of indicated purified proteins, followed by immunoblotting.

Comment: 5. Although the authors show that RBBP7 depletion in cells reduces BUB3 ubiquitination (Fig. 4f), this is not evidence that CUL4-RBBP7 itself ubiquitinates BUB3. The same results were observed for RepID (Fig. 3e) but the authors are not concluding that RepID ubiquitinates BUB3.

Comment: 6. The major concern is the absence of evidence that CRL4-RBBP7 ubiquitinates BUB3 in vitro. For this model to be convincing the authors should show that reconstituted CRL4-RBBP7 ubiquitinates BUB3.

7. In addition since the authors claim that CRL4-RepID does not ubiquitinate BUB3, even though RepID interacts with BUB3 (whether this is direct or indirect was not shown), the activity of CRL4-RepID towards BUB3 should also be tested.

Response: We thank the reviewer for raising these important points. To assess if RBBP7 directly ubiquitinates BUB3, we performed additional in vitro experiments (new Supplementary Fig. 3i, left). Our data show that the reconstituted CRL4-RBBP7, but not CRL4-RepID, can ubiquitinate BUB3. These data suggest that the presence of RBBP7 in the reconstituted complex being critical for BUB3 ubiquitination. RepID, which is required to recruit CUL4 to chromatin, was required for BUB3 ubiquitination in cells but was dispensable in vitro.

Supplementary Fig. 3i: In vitro ubiquitination assay using purified proteins, followed by immunoblotting with BUB3 antibody.

Comment: 8. The authors appear to assume that is kinetochore-associated BUB3 that is degraded by CRL4-RBBP7. However have they considered BUB3 in the complex with the mitotic checkpoint complex?

Response: Yes, thank you, we agree with the reviewer that RepID-induced BUB3 degradation could occur either in the context of the kinetochore or within the mitotic checkpoint complex. We have modified the discussion (page 14, second paragraph) and our model (Fig. 7) to emphasize this point.

Comment: 9. Lines 67-69. The mechanism of SAC silencing is incorrect. The APC/C is activated because the MCC dissociates from the APC/C-Cdc20 complex due to Cdc20 (of the MCC) of BubR1 ubiquitination.

Response: We thank the reviewer for pointing out this issue. We changed the text according to the reviewer's suggestion.

In conclusion, we would like to thank all the reviewers again for their insightful comments and suggestions. We believe that these suggestions helped improve the manuscript and helped strengthen the conclusions, and we are grateful for the opportunity to submit this revision for your consideration.

REVIEWERS' COMMENTS:

Reviewer #1 (Remarks to the Author):

I would like to thank the authors for great work during revision. I have only two minor suggestions:

1. Line 271 the word should be "viability"
2. In terms of data shown in Sup Fig 3b, in the absence of RepID, RBBP7 binding to CUL4 and DDB1 is reduced. To me this strongly supports that RepID HELPs the "handover" of RBB7 to CUL4. This was not mentioned in the text.

Reviewer #2 (Remarks to the Author):

The authors have performed new experiments and adequately answered the questions raised in my review comments. The manuscript is improved.

Reviewer #3 (Remarks to the Author):

The authors have performed extensive new experiments that have satisfactorily addressed my concerns and I am happy to recommend this interesting paper for publications in Nat Comms.

Responses to the reviewers' comments:

Reviewer #1 (Remarks to the Author):

I would like to thank the authors for great work during revision. I have only two minor suggestions:

1. Line 271 the word should be "viability"
2. In terms of data shown in Sup Fig 3b, in the absence of RepID, RBBP7 binding to CUL4 and DDB1 is reduced. To me this strongly supports that RepID HELPs the "handover" of RBBP7 to CUL4. This was not mentioned in the text.

Response: Thank you for this evaluation and for the suggestions. We have corrected the typo in line 271 and have inserted a sentence in the Results section suggesting that RepID helps the handover, as suggested.

Reviewer #2 (Remarks to the Author):

The authors have performed new experiments and adequately answered the questions raised in my review comments. The manuscript is improved.

Response: Thank you for this evaluation.

Reviewer #3 (Remarks to the Author):

The authors have performed extensive new experiments that have satisfactorily addressed my concerns and I am happy to recommend this interesting paper for publications in Nat Comms.

Response: Thank you for the evaluation.

In closing we would like to thank all the reviewers for their helpful suggestions in the first round of review, which notably improved the submission, and for re-reading and evaluating the revised version. We are looking forward to see the paper published in Nature Communications.